# Equatorial mountains on Pluto are covered by methane frosts resulting from a unique atmospheric process

Tanguy Bertrand [1,2✉], François Forget [2✉], Bernard Schmitt [3], Oliver L. White[1,4] & William M. Grundy [5]

Pluto is covered by numerous deposits of methane, either diluted in nitrogen or as methane-rich ice. Within the dark equatorial region of Cthulhu, bright frost containing methane is observed coating crater rims and walls as well as mountain tops, providing spectacular resemblance to terrestrial snow-capped mountain chains. However, the origin of these deposits remained enigmatic. Here we report that they are composed of methane-rich ice. We use high-resolution numerical simulations of Pluto's climate to show that the processes forming them are likely to be completely different to those forming high-altitude snowpack on Earth. The methane deposits may not result from adiabatic cooling in upwardly moving air like on our planet, but from a circulation-induced enrichment of gaseous methane a few kilometres above Pluto's plains that favours methane condensation at mountain summits. This process could have shaped other methane reservoirs on Pluto and help explain the appearance of the bladed terrain of Tartarus Dorsa.

[1] National Aeronautics and Space Administration (NASA), Ames Research Center, Space Science Division, Moffett Field, CA 94035, USA. [2] Laboratoire de Météorologie Dynamique, IPSL, Sorbonne Universités, UPMC Université Paris 06, CNRS, BP99, 4 place Jussieu, 75005 Paris, France. [3] Université Grenoble Alpes, CNRS, Institut de Planétologie et d'Astrophysique de Grenoble, 38000 Grenoble, France. [4] The SETI Institute, Mountain View, CA 94043, USA. [5] Lowell Observatory, Flagstaff, AZ 86001, USA. ✉email: tanguy.bertrand@nasa.gov; forget@lmd.jussieu.fr

An important observation of Pluto made by the New Horizons spacecraft in July 2015 was the great geomorphological diversity and albedo contrast of terrains in its equatorial regions[1–6]. In particular, west of Sputnik Planitia, the region of Cthulhu is characterized by a spectacular dark mantling that has been interpreted as an accumulation of haze particles that have settled from the atmosphere[7], or in some locations a cryovolcanic deposit[8].

Although most of Cthulhu's surface appears volatile-free[3], the Long-Range Reconnaissance Imager (LORRI) instrument onboard New Horizons revealed the presence of patchy bright deposits at specific locations[6] and the Multispectral Visible Imaging Camera (MVIC) showed that they contain methane ($CH_4$)[2]. Figure 1a, b shows a part of eastern Cthulhu containing isolated high-altitude mountain chains, known as Pigafetta Montes, the crests of which reach 2.5–3.5 km above their bases and almost 4 km above mean radius[9]. They are capped with bright frosts (with an albedo of ~0.65[10]) above 1.5 km altitude with a striking resemblance to terrestrial alpine landscapes

(Fig. 1c). MVIC showed that these frosts contain $CH_4$ but could not conclude on its state ($CH_4$-rich ice, $CH_4$ diluted in $N_2$-rich ice or both)[2,11].

Why does $CH_4$ ice form on top of these mountains? It has been suggested that the sublimation and condensation of volatile ices could drive the ices out of thermodynamic equilibrium and result in altitude segregation with $N_2$-rich ice dominating at low-elevations and $CH_4$-rich ice dominating at high elevations[12,13]. Here we explore an alternative scenario that involves an atmospheric process. On Earth, atmospheric temperatures decrease with altitude, mostly because of adiabatic cooling and warming in upward and downward air motions, respectively. As a consequence, surface temperatures also decrease with altitude because the surface is cooled by the dense atmosphere through sensible heat fluxes. In such conditions, as moist wind approaches a mountain, it rises upslope and cools adiabatically, leading to condensation and formation of snow on top of mountains. Note that here frost is defined as ice crystals that form directly on a below-freezing surface via a phase change from gas in the

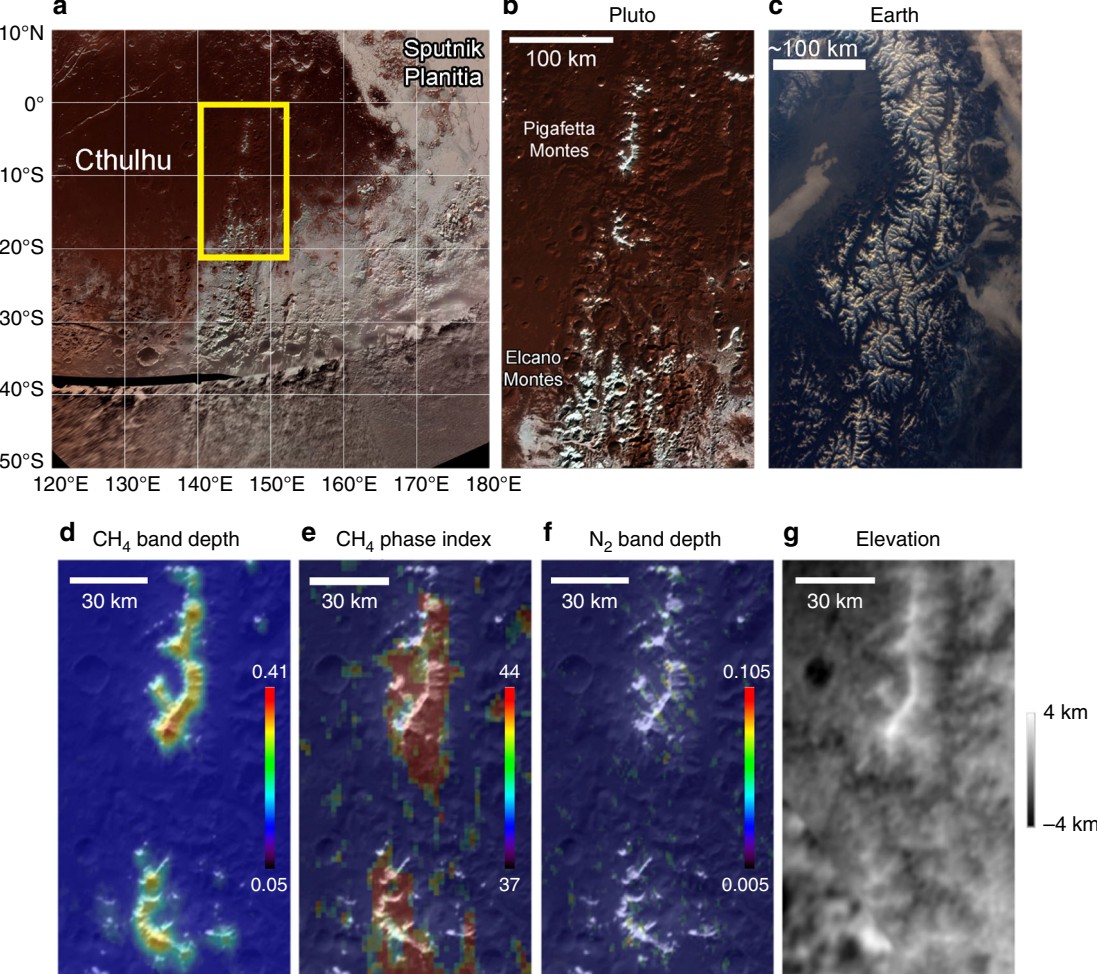

**Fig. 1 Detection of CH₄-rich ice on top of Pigafetta and Elcano Montes. a** New Horizons map of the southeast Cthulhu region on Pluto, located in the equatorial regions west of Sputnik Planitia. Yellow box indicates the boundaries of the area seen in detail in (**b**). **b** Detail of the $CH_4$ frost-capped ridges of Pigafetta and Elcano Montes within Cthulhu Macula (148.2°E, 10.1°S), seen in an enhanced Ralph/MVIC colour image (680 m/pixel, cylindrical projection). **c** Satellite view of water ice-capped mountain chains in the Alps. **d** LEISA $CH_4$ band depth map focusing on the Pigafetta Montes within Cthulhu, superimposed on the visible map, with blue-to-red indicating increasing $CH_4$ absorption. **e** $CH_4$ phase index map (at ~3× lower spatial resolution, see definition in 'Methods', Phase index maps), with red corresponding to $CH_4$-rich ice and green-blue to $N_2$-rich ice containing $CH_4$. Yellow corresponds to some mixture of both phases. The high value of the $CH_4$-phase index likely indicates that only the $CH_4$-rich phase is present. **f** Same as (**d**) for the $N_2$ band depth map. **g** New Horizons elevation map. Lateral resolution of the topography data at Pigafetta Montes is ~2.5 km, with a vertical precision of 230 m[11].

atmosphere, whereas snow is defined as individual ice crystals that grow while suspended in the atmosphere and subsequently fall as precipitation onto the surface.

Could this process apply to Pluto too? Both observations and modelling show that, unlike on Earth, there is a strong increase in atmospheric temperatures with altitude in the first kilometres above the surface[14,15] because of the heating resulting from the absorption of solar radiation by $CH_4$ gas (except when $N_2$ ice is present on the surface, because local $N_2$ ice sublimation can cool the lowermost few kilometres of the atmosphere[14,15]). The atmosphere is too thin to affect the surface temperature itself and in the absence of $N_2$ ice, the surface remains in local radiative balance everywhere, independent of the altitude and colder than the atmosphere above. One consequence of this is that the near-surface air is cooled and tends to flow downslope because it is denser than the air away from the slope at the same level. Climate simulations confirm this trend and indicate that these katabatic downslope near-surface winds dominate everywhere and at all times of day on Pluto[14]. Under such conditions, it is impossible to explain the condensation of $CH_4$ by upward air motion as on the Earth, and a different mechanism must be identified.

Here we demonstrate that the bright frosts observed in Cthulhu are mostly made of $CH_4$-rich ice. We then use a numerical climate model of Pluto to investigate the origin of their formation. Our simulations reproduce the accumulation high-altitude $CH_4$ ice where the frost-capped mountains are observed, in particular on the ridges and crests of the Pigafetta and Elcano Montes in eastern Cthulhu. They show that $CH_4$ condensation is favoured by sublimation-induced circulation cells that seasonally enrich the atmosphere with gaseous methane at those higher altitudes.

## Results

**Detection of $CH_4$-rich ice on the top of the mountain chains.** Figure 1d–g shows $CH_4$ and nitrogen ($N_2$) band depth and $CH_4$-phase index maps[3] derived from the highest spatial resolution spectral-image of the Linear Etalon Imaging Spectral Array (LEISA) instrument, presented alongside the latest digital elevation model[9]. These maps indicate that the bright frosts on the top of the mountain chain are mostly composed of $CH_4$-rich ice. Only a few very small $N_2$-rich ice patches occur in this area, mostly located at the bases of these mountains and in the valleys separating them.

Another geographical context where $CH_4$ ice was detected was on north-facing walls and rims of many craters in Cthulhu, including Edgeworth Crater and many smaller examples surrounding Pigafetta Montes. The composition and phase maps also show that these deposits are $CH_4$-rich ice[3]. We can straightforwardly explain this behaviour by calculating that these slopes received less insolation than the south-facing slopes during the previous northern fall and winter (see Supplementary Figs. 8, 9). They could thus have acted as cold traps for $CH_4$ ice.

**Numerical climate simulations of Pluto.** Here, we use the Pluto Global Climate Model (GCM) of the Laboratoire de Météorologie Dynamique (LMD), designed to simulate the climate and $CH_4$ cycle on Pluto[14,16] (see 'Methods', The LMD Pluto GCM). It includes a full description of the volatile exchange between surface ice and atmospheric gas (for $N_2$, $CH_4$ and CO), transport and turbulent mixing of gaseous $CH_4$ in the atmosphere and an implicit scheme for the formation of $CH_4$ clouds. The radiative effect of $CH_4$ and CO is also included[14]. The model provides an evolution of surface pressure and of the $CH_4$ and CO mixing ratio in good agreement with New Horizons and terrestrial observations[14,17]. In

the model, we rely on Raoult's law as a substitute for the ternary equation of state. We consider that $CH_4$-rich ice behaves like pure $CH_4$ and that $N_2$-rich ice contains 0.5% of $CH_4$ (see 'Methods', $CH_4$ and CO condensation–sublimation on the surface). We note that this approximation can lead to some uncertainties in $CH_4$ solid-phase stability but that it remains a reasonable approach for the focus of this paper ($CH_4$-rich deposits) and does not qualitatively change the results of this paper.

To simulate Pluto as observed in 2015, we performed a simulation similar to the one presented in Bertrand et al.[16] but with the following improvements. Firstly, we used the latest topography data of the encounter hemisphere of Pluto[9], which includes Sputnik Planitia, eastern Cthulhu, and Tartarus Dorsa (the latter being the western extent of the bladed terrain, a chain of massive, low-latitude, high-elevation deposits of $CH_4$ ice that have been eroded via sublimation into the eponymous blades, and which extend across much of the sub-Charon side of Pluto[11,18]). Secondly, perennial $CH_4$ deposits were added on the sub-Charon side of Pluto (covered by low-resolution New Horizons imaging) wherever terrain with diagnostic characteristics of the bladed terrain was detected[18] (i.e. intermediate albedo in LORRI approach imaging, strong $CH_4$ spectral signature in MVIC imaging, high elevation in far side limb profiles). Thirdly the initial state of the simulation was obtained for Earth year 1984 from a 30-million-year simulation performed with the Pluto volatile transport model[19,20] (see 'Methods', Initial state of the reference simulation and grid resolution), thus allowing a steady state for ice distribution, surface and soil temperatures to be reached. We used two horizontal resolutions: a baseline resolution of 7.5° in latitude, 11.25° in longitude (i.e. ~150 km) for the years 1984–2014 and a higher resolution of 2.5° in latitude and 3.75° in longitude (i.e. ~50 km) for the years 2014–2015. This allowed us to better represent the atmosphere-topographic interactions during the New Horizons encounter.

**Net diurnal $CH_4$ deposition in Cthulhu in 2015.** In our simulation, the $CH_4$ atmospheric mixing ratio reaches 0.4–0.7% in 2015, with slightly higher values in the northern hemisphere due to increasing $CH_4$ ice sublimation from the mid-to-high northern latitude plains, as predicted by previous climate simulations[14]. A net $CH_4$ ice deposition of ~20 μm over the year 2015 is obtained locally in Cthulhu, at the summits and on the flanks of the highest mountains of Pigafetta and Elcano Montes, in qualitative agreement with New Horizons observations (Fig. 2a). In this first simulation, $CH_4$ ice affects the surface albedo by increasing it from 0.1 (volatile-free, or deposits thinner than 1 μm) to 0.5 (roughly the albedo of the $CH_4$-rich bladed terrain, for deposits thicker than 1 μm). However, on Pluto, it is likely that once a fresh initial $CH_4$ frost grows to a few millimetres, the surface albedo increases to values even higher than 0.5, inducing a stronger cooling (reduced solar heating) of the surface and allowing more $CH_4$ to condense there. This positive feedback should favour even stronger katabatic winds and the accumulation of more ice on top of Pluto's mountains. To explore its effect, we performed a second simulation in which the $CH_4$ ice–albedo was slowly increased up to 0.9 as soon as the ice thickness reached more than 1 μm (see 'Methods', Surface albedo feedback for equatorial $CH_4$ ice). In this second simulation, the modelled $CH_4$ condensation rates are stronger than in the first, in particular in the diurnal zone above ~20°S with a net $CH_4$ ice deposition over the year 2015 of ~40 μm and on top of Pigafetta Montes and a better agreement with New Horizons observations (Fig. 2b). By extrapolating over the last 27 Earth years (beginning of northern spring), such an accumulation would have covered the tops of

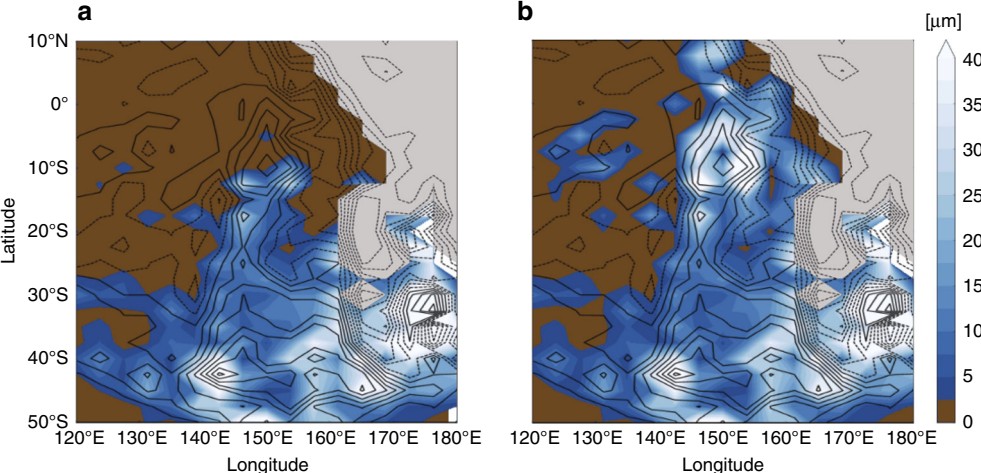

**Fig. 2 Modelled deposition of CH₄ ice in eastern Cthulhu. a** Net surface CH₄ ice accumulation (from blue to white) in Cthulhu (covering the same area as in Fig. 1a) obtained in our simulation for 2015. Superimposed topography contours are at 300-m intervals. The volatile-free surface is shaded in brown, the Sputnik Planitia N₂ ice sheet is shaded in grey. **b** The same as (**a**) but using an amplifying albedo feedback for CH₄ ice (see 'Methods', Surface albedo feedback for equatorial CH₄ ice). The albedo feedback has no effect below ~20°S because these latitudes receive little to no insolation (polar night) in 2015.

these mountains with a ~1-mm-thick CH₄ ice deposit. However, the frosts could have started forming during northern fall or winter and accumulated to thicknesses much >1 mm. It may also be possible that they grew thicker (up to a few metres) over multi-annual timescales, during past climate epochs with larger amounts of gaseous CH₄ available for condensation[20]. The lack of image resolution, the uncertainties on the starting season of the frosts and the model approximations for computing CH₄ condensation and sublimation rates (see 'Methods', CH₄ and CO condensation–sublimation on the surface) prevent us for estimating the thickness of CH₄-rich deposits with great confidence. Nevertheless, the 2015 simulation sheds light on the possible processes forming the CH₄-rich deposits seen at mountain summits in eastern Cthulhu.

**What drives the formation of high-altitude CH₄-rich frosts?** Condensation on the surface occurs when the near-surface atmospheric CH₄ mixing ratio is greater than the CH₄ mixing ratio at saturation (see 'Methods', CH₄ and CO condensation–sublimation on the surface). As mentioned above, on a N₂-free surface such as in Cthulhu, there is no dependence of surface temperature on altitude (Supplementary Fig. 1a, c). The CH₄ mixing ratio at saturation, which depends only on surface temperature, is therefore independent of altitude and should be relatively constant for given surface properties and insolation. Our model indicates that despite their very low albedo (~0.1), the equatorial regions in Cthulhu in 2014–2015 are cold enough during night-time (~40–42 K) to trigger CH₄ condensation onto the surface and the formation of μm-thin CH₄ frosts (Fig. 3, Supplementary Fig. 1a, b). In the model, these frosts then entirely sublime during daytime, when the surface heats up to ~45–48 K (Fig. 3c, Supplementary Fig. 1c, d), except over high-altitude terrains, where the night-time condensation dominates the day-time sublimation (Supplementary Fig. 2), allowing the frosts to subsist and grow thicker. We determine from the analysis of our results that this is because the near-surface atmospheric CH₄ mixing ratio in the model strongly varies with altitude in the equatorial regions, with an enrichment in gaseous CH₄ above ~4 km altitude (above mean radius, Supplementary Fig. 3) and a

depletion in the lowest levels of the atmosphere. This vertical distribution of gaseous CH₄ in the first kilometres above the surface forms self-consistently as an outcome of our GCM simulation but remains unconstrained by observations as the CH₄ mixing ratio was not observed by New Horizons below 80 km altitude. As a result, the modelled near-surface atmospheric CH₄ mixing ratio and therefore the condensation rates are higher on top of the mountains (which peak into an atmosphere richer in CH₄) than in the depressions (Fig. 3a, b). As shown in 'Methods' (see CH₄ and CO condensation–sublimation on the surface), the CH₄ condensation is also controlled by the near-surface winds, which mix gaseous CH₄ near the surface (turbulent mixing). The model predicts that on Pluto, these winds tend to be stronger on the steepest slopes, such as on the flanks of Pigafetta Montes where slopes can range up to 45°[9] (Fig. 3a), which enhances CH₄ condensation rates there.

## Discussion
Why is Pluto's equatorial atmosphere depleted in gaseous CH₄ near the surface and enriched at higher altitudes in the model? This could have been a consequence of CH₄ cloud formation, with condensation in the cold, lowest atmospheric levels, but we ruled out this hypothesis by performing a simulation in which atmospheric condensation was artificially neglected: it yielded the same results.

In fact, the model indicates that ascending winds in the western regions of Sputnik Planitia are responsible for this enrichment by transporting gaseous CH₄ upward. Analysing the general circulation indicates that two different mechanisms are involved. First, as N₂ sublimes in the north and condenses in the south, it triggers a meridional circulation dominated by a north-to-south flow and characterized by one circulation cell centred above Sputnik Planitia[16]. As CH₄ ice sublimes in the northern hemisphere, it is transported southward and toward higher altitudes above the equatorial regions by this circulation cell. Second, the interaction between the zonal flow and the high western boundary of Sputnik Planitia's basin produces vertical upward motions which also contributes to the vertical transport of gaseous CH₄.

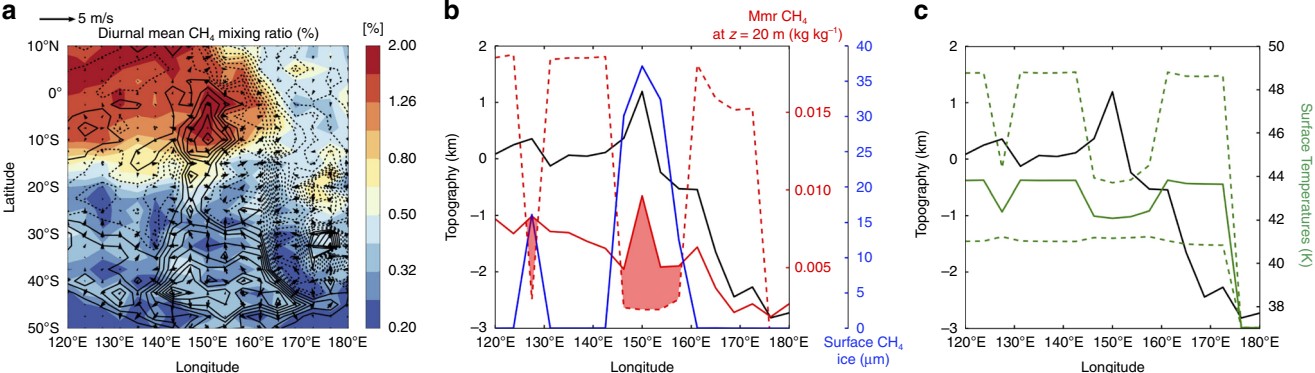

**Fig. 3 Atmospheric CH₄ abundance and surface temperatures drive CH₄ condensation and accumulation on mountain summits. a** Near-surface CH₄ atmospheric mixing ratio in Cthulhu as simulated by our model (filled contours, in %) with arrows indicating winds at 5 m above the surface. **b** Cross-section of Cthulhu at 5°S showing the diurnal mean CH₄ mass mixing ratio (kg kg⁻¹) above the local surface (red solid line) and the diurnal mean CH₄ mass mixing ratio at saturation (red dotted line), as obtained in our simulation for July 2015; the topography; and the thickness of the surface layer of CH₄ ice (black solid line); and the thickness of the surface layer of CH₄ ice (about ~40 μm on top of Pigafetta Montes, blue solid line). The shaded areas indicate where there is saturation of CH₄ above the surface (in diurnal mean) that leads to condensation onto the surface. **c** Same cross-section showing the diurnal mean surface temperature (solid line) as obtained in our simulation. The diurnal minimal and maximal surface temperatures are also indicated (dotted lines). Colder daytime temperatures are obtained where CH₄ ice is present, due to the brighter surface (albedo positive feedback). Maps are shown in the Supplementary Figs. 1 and 2.

Overall, the formation of CH₄ frost on top of Pluto's mountains appears to be driven by a process completely different from the one forming snow-capped mountains on the Earth, according to our model. It is remarkable that two phenomena and two materials that are so dissimilar could produce the same landscape, when seen at similar resolution (Supplementary Fig. 4). The proposed plutonian process creating frost-capped mountains may control an important factor governing the CH₄ cycle on Pluto in general. In particular, it might help explain the bladed texture of Tartarus Dorsa, the massive CH₄ deposits located east of Sputnik Planitia[11], by favouring CH₄ condensation at higher altitude (bladed terrain is found almost exclusively at elevations >2 km above the mean radius[8]). In such conditions, initially elevated portions of the CH₄ deposits may represent sites of increased CH₄ condensation relative to lower portions, and so their positive topographic relief may gradually be amplified. This would complement the 'penitente model'[11,21] in which the bladed texture is suggested to be mostly induced by sublimation processes.

## Methods

**The LMD Pluto GCM**. We used the LMD Pluto GCM[14,16] to simulate Pluto's climate and the methane cycle in 2015. The model includes atmospheric dynamics and transport, turbulence, radiative transfer and molecular conduction as well as phase changes for N₂, CH₄ and CO. The GCM reproduces well the thermal structure measured by the New Horizons spacecraft in the lower atmosphere (below 200 km altitude) and the threefold increase in surface pressure observed from stellar occultations between 1988 and 2015 with ~1.1 Pa in 2015[5,22]. Recent improvements to the model include incorporation of perennial high-altitude CH₄ deposits in the equatorial regions (bladed terrain), based on mapping of Pluto's far side[18], and the use of the latest topography from New Horizons data[9]. We use flat topography for the non-observed southern hemisphere. Adding topography in the southern hemisphere does not impact the results of this paper. The general circulation is strongly controlled by the north-to-south N₂ flow and is not significantly impacted by the presence of N₂ ice deposits outside Sputnik Planitia[16].

**Initial state of the reference simulation and grid resolution**. The initial state for the GCM is derived from simulations performed with the 2D LMD Pluto volatile transport model over 30-million years, taking into account the orbital and obliquity changes of Pluto over time[16,19,20]. These long-term simulations allow the surface ice distribution (N₂, CH₄ and CO ices), surface temperatures and soil temperatures to reach a steady state for current-day Pluto. The initial state of the atmosphere in the 3D GCM is an isothermal profile. We run the GCM from 1984 to 2015 which is sufficient to reach a steady state for the atmosphere and a realistic circulation regime insensitive to the initial state[14].

The long-term volatile transport simulations and the low-resolution GCM simulations are performed with a horizontal grid of 32 × 24 points to cover the globe (i.e. 11.25 × 7.5°, ~150 km in latitude) and 27 vertical levels. Simulations for the years 2014 and 2015 are then performed with a higher spatial resolution by using a grid of 96 × 72 points (3.75 × 2.5, ~50 km in latitude) and 47 vertical levels (with a first level at $z_1 = 5$ m).

**Surface conditions**. Our GCM simulations have been performed using an N₂ ice emissivity of 0.8 and albedo of 0.7. The surface N₂ pressure simulated in the model is constrained by these values. The albedo and emissivity of the bare ground (volatile-free surface) are set to 0.1 and 1, respectively, which corresponds to a terrain covered by dark materials such as Cthulhu[10]. Methane ice emissivity is fixed to 0.8 in all simulations. We use a CH₄ ice–albedo of 0.5 for the equatorial deposits (except when the albedo feedback scheme is on, see below) and of 0.65 for the polar deposits, which is consistent with the available albedo maps of Pluto[10].

The thermal conduction into the subsurface is performed with a low thermal inertia near the surface, set to $I_d = 20$ J s⁻⁰·⁵ m⁻² K⁻¹, to capture the short-period diurnal thermal waves and a larger thermal inertia below set to $I_s = 800$ J s⁻⁰·⁵ m⁻² K⁻¹ to capture the much longer seasonal thermal waves which can penetrate deep into the high-TI substrate. The rest of the settings are similar to the previous simulations of Pluto with the LMD GCM[14,16]. We assume a density for CH₄ ice of 500 kg m⁻³ [20].

**CH₄ and CO condensation–sublimation on the surface**. CH₄ and CO are minor constituents of Pluto's N₂ atmosphere and their surface–atmosphere interactions depend on the turbulent fluxes given by:

$$F = \rho\, C_d U\, (q - q_{surf}) \qquad (1)$$

with $q_{surf}$ the saturation vapour pressure mass mixing ratio (in kg kg⁻¹) at the considered surface temperature, $q$ the atmospheric mass mixing ratio, $\rho$ the air density, $U$ the horizontal wind velocity and $C_d$ the drag coefficient at 5 m above the local surface ($C_d = 0.06$). $q_{surf}$ is computed using the thermodynamic Claudius–Clapeyron relation[23] for CH₄ with a latent heat for sublimation of 586.7 kJ kg⁻¹.

On Pluto's surface, the volatile ices should form solid solutions whose phases follow ternary phase equilibria; they do not exhibit ideal behaviour[12,24]. We note that sophisticated equations of state exist for the N₂–CH₄ and N₂–CH₄–CO systems under Pluto surface conditions (CRYOCHEM[12]). These ternary and binary systems, as currently understood[12], are able to explain a great diversity of phases (CH₄-rich, N₂-rich and N₂-rich+CH₄-rich solids) that are seen on Pluto within the range of expected temperatures and relatively unvarying surface pressure and strongly N₂-dominated vapour composition seen on Pluto.

At the temperature of the CH₄ deposits modelled in this paper (~45 K, prevailing at high altitudes), the ternary phase equilibria shown by Tan and Kargel[12] predict that two phases coexist: a very nearly pure CH₄ solid (≪1% impurities of N₂ and CO in solid solution) and N₂-rich vapour. This is consistent, to first order, with the observations by New Horizons and with what we report in this paper and in Fig. 1.

However, as these equations of state have not been coded for use in a GCM or been applied to the specific distribution of ices and temperatures seen on Pluto or in a GCM, we have substituted the alternative of relying on Raoult's law, as described next. It is possible that for conditions of rapid frost deposition, the solid condensates may be amorphous mixtures, which might tend to exhibit thermodynamics somewhat like those implied by Raoult's law. Accordingly, we interpret Pluto to be a non-equilibrium dynamical environment with continuous exchange of materials (condensation, sublimation, atmospheric transport…), including on daily timescales where departures from equilibrium could be likely.

Future work involving laboratory experiments, spectroscopic analyses, thermodynamic models and GCMs is strongly needed to improve the models, constrain the timescales for ice relaxation toward thermodynamics equilibrium and explore in detail the effect of the ternary phase equilibrium on Pluto (and on other Trans-Neptunian objects).

Here, for simplicity in coding with a GCM, we have adopted Raoult's law, as in previous GCM studies[14,16,17]. We consider the mixtures $N_2$:$CH_4$ and $N_2$:CO with 0.5% of $CH_4$ and 0.3% of CO respectively, as retrieved from telescopic observations and from New Horizons observations[4,25]. However, we note that Raoult's law was at first intended for vapour–liquid equilibria and ideal solutions only, which do not include the $N_2$–$CH_4$–CO system observed on Pluto. Despite the fact that this approximation gives good results and allows for reproducing the atmospheric mixing ratios observed by New Horizons observations[17], it may still introduce errors on sublimation and condensation rates of the different types of ice. This approximation seems sufficient for our present needs and for the study of $CH_4$-rich ice deposits on Pluto's surface, which are the focus of this paper, but we acknowledge that it leads to some unevaluated uncertainties in $CH_4$ solid-phase stability.

**Surface albedo feedback for equatorial $CH_4$ ice**. We tested the impact of an amplifying surface ice–albedo feedback on the results. To do that, we allowed the surface $CH_4$ ice–albedo to change depending on the thickness of ice $Q$ present on the surface. We assumed a minimal albedo of $A_{min} = 0.1$ for deposits thinner than $Q_{lim} = 1\ \mu m$, and a maximal albedo of $A_{max} = 0.9$ for the thickest deposits. In the model, this change is applied to $CH_4$ ice in the equatorial regions only, outside the bladed terrain deposits (i.e. only recent frost). We use a simple hyperbolic tangent function for the transition between the extreme albedo values (tanh functions are commonly used in climate models to represent time-variation of icy surfaces albedo[26]):

$$\frac{A_{max} - A_{min}}{2} \tanh\left(\frac{Q - Q_{lim}}{2} - 2\right) + A_{max} - A_{min} \qquad (2)$$

In our model, this ice–albedo scheme leads to a strong positive feedback when $CH_4$ condenses on the surface, as it lowers the surface temperature and therefore the near-surface $CH_4$ mixing ratio at saturation, thus allowing for a further increase of the $CH_4$ condensation rate. Note that the goal here is to boost the effect of albedo feedback with a simple representation within the time of the simulation, but these changes in surface albedo may actually occur over longer timescales, and be efficient for deposits thicker than ~1 mm.

**Phase index maps**. The phase index, as defined in Schmitt et al.[3] is an index allowing discrimination between $CH_4$ diluted in $N_2$ ice and $CH_4$-rich ice phases based on the position of a set of $CH_4$ near-infrared bands. It is thus based on the measure of the shift of these bands upon dilution in $N_2$ ice as measured in the laboratory[27].

At the spatial resolution of the LEISA measurements (2.7 km in the high-resolution strip used for Fig. 1d, f, and 7 km used for the 1e panel) we cannot discriminate between a patchwork distribution of $N_2$-rich and $CH_4$-rich ice at scale below a few kilometres, with an intimate mixture of the crystals of both phases or a vertical stratification (but at sub-mm scale). However, taking into account the noise and detection level, the maps show that nitrogen may be only present in some very localized area at low altitude. The phase index maps clearly point to a $CH_4$-rich dominant composition.

**Reporting summary**. Further information on research design is available in the Nature Research Reporting Summary linked to this article.

## Data availability
The GCM data is freely available from the corresponding authors upon request.

## Code availability
All model versions are freely available upon request by contacting the corresponding authors.

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

## Acknowledgements
We thank Kevin Zahnle, Jeff Moore, John Wilson, Bob Haberle and Melinda Kahre for insightful discussions and stimulating discussions and comparisons between Mars and Pluto. We thank the NASA New Horizons team for their excellent work on a fantastic mission and their interest in this research. We thank the Centre National d'Etudes Spatiales (CNES) for its financial support through its "Système Solaire" programme. T.B. was supported for this research by an appointment to the National Aeronautics and Space Administration (NASA) Post-doctoral Programme at the Ames Research Center administered by Universities Space Research Association (USRA) through a contract with NASA.

## Author contributions
T.B. and F.F. designed and developed the model. T.B. performed the simulations. T.B. and F.F. made the modelling figures, B.S. and W.G. produced the LEISA maps, T.B. and F.F. wrote the manuscript with significant contributions from B.S. and O.W.

## Competing interests
The authors declare no competing interests.
