## [Peer Review File · Nature Communications]

REVIEWER COMMENTS

Reviewer #1 (Remarks to the Author):

This paper reports the most recent results of an ongoing line of investigation into processes affecting Pluto's seasonal evolution based on observations obtained during the 2015 flyby by New Horizons and a Pluto global circulation model developed and periodically extended by the leading authors since then. The focus of the present paper is on the interpretation of

Pluto's methane-rich surface ice deposits, including their origin on mountain tops and crater walls, ridges and crests, and the formation of the bladed terrain. Processes forming these deposits are investigated using numerical simulations of Pluto's climate that include their existing model of the methane cycle. Based on these simulations, a process is proposed for forming the high-altitude deposits that is quite different from the adiabatic cooling of air uplifted by terrestrial mountains. Instead, mountain top/ridge methane condensation would be favored by sublimation-induced circulation cells that super-seasonally enrich atmospheric methane at these higher altitudes.

Pluto science has a high level of visibility and community interest owing to the rich store of data returned by that mission that is still being analyzed. These results will therefore be of wide interest to a multidisciplinary science community. Also included in this paper and the Supplementary Discussion are several highly-enhanced data products from the mission, with color stretched for clarity, showing unprecedented detail in the topographic distribution of methane and a red material. These figures will be of special interest to the readership. In particular, they support the modeling and the identification of the mountain top deposits as methane-rich ice. The authors suggest the red material may be ancient, highly processed, methane that has been exposed over super seasons.

However, the interpretation - or perhaps wording - in the text sometimes confuses a successful simulation with reality. Sometimes a simulated process is treated as a candidate or proposed hypotheses, and the result is properly evaluated in terms of consistency with the observations, with a good fit leading to the basis for making a prediction for further testing; e.g., lines 130-160 in the MS Word document. Otherwise a simulation result in agreement with observation is treated as a successful test of an included hypothesized process; or a process discovered in a simulation that successfully fits the observations is sometimes held up to be actually occurring on Pluto, rather than being a promising candidate. Such is the case for the process proposed for depositing frost at high altitudes. The Abstract asserts that processes on Pluto forming the high-altitude deposits are completely different than on Earth, do not result adiabatically, but result from a circulation cell. In verifiable reality, these are predictions of the model, as confirmed on line 73. This process may well be correct. But these simulations do not prove that. No evidence is given that these simulations uniquely represent the observational data set. Another model with a different driving hypothesis

might equally fit the data. Results from simulations are model-dependent and their validity depends on the completeness and uniqueness of the model.

A particular pathway through which the employed model may prove to be non-unique, or even wanting, is the simplification (given in the Methods section) of using Raoult's law to treat condensation-sublimation of the CH₄:N₂:CO solid solution, rather than using the more complicated ternary phase diagram. This would require knowledge of the equation of state that is not generally available, and such theoretical ternary diagrams are not yet experimentally verified.

Also, regarding the issue of uniqueness, Tan & Kargel (2018) have previously proposed a competing process for explaining the deposition of the methane-rich solid solution on mountain tops/ridges that is based on their theoretical model of the equation of state. They show that local deviations from thermal equilibrium can sensitively result in the formation of methane-rich or nitrogen-rich ice according to the circumstance of the deviation. Their Fig. 2b, for Pluto, indicates the transformation of the nitrogen-rich phase VS_1 at low methane mole fractions to the methane rich phase VS_2 at ambient isobaric temperatures higher than for phase equilibrium. This would occur at high elevations, where the atmosphere is warmer than the isobaric condensation temperature.

Although Tan & Kargel have been referenced (on lines 334-335), the reference is only in the context of the ternary diagram, not in terms of their proposed explanation for the deposition on mountain ridges (among other Pluto ice phase phenomena).

I do not recommend publication until these issues have been addressed satisfactorily. However, I believe there is enough new science and potential interest in this paper that a revised version should then merit publication in Nature. At a minimum, this paper should acknowledge and compare to the previous work, and place the proposed cellular deposition process in the context of an alternative model. The revised text and affected results of the fits should frame the conclusions as conditional, as has properly been done elsewhere in the text. This would be facilitated by making the suggested substitutions below.

Minor comments/corrections:

Define "phase-index" (as in maps).

Line 114 – "CH₄ gas"?

115 – "..., which suppresses rapid methane sublimation."

- 115 – “...local radiative balance everywhere.”
- 119 – replace “show” by “indicate” or “suggest”
- 120- replace “it is” by “It would be”
- 121- replace “must” by “would have to be”
- 166 - replace “shows” by “indicates” or “suggests”
- 172 – ...mixing ratio in the model...
- 176 – replace “We also found...” by “The model also predicts...”
- 185 – replace “shows” by “indicates”
- 194, - put comma after “Overall”
- 196 – replace “The “plutonian” process...” by “The proposed “plutonian” process...”
- 200 – replace “will” by “may”
- 201 - replace “will” by “may”

Figures

- 262 – replace “Same as A...” by “Same as D...”
- 265 – replace “than” by “as”
- 349 – append to sentence: “ ..., as the temperature is lowered.”

References on page 15 do not continue the numbering of page 9.

List “20 to 25” instead of “18, 19, 20, 21, 22, 24”

Supplementary Discussion

- Line 47 – Define “spectral index”
- 60 – replace “because of “ by “which we attribute to” or by “which the modeling attributes to”
- 75 – replace “receive” by “received”
- 103 – replace “indicate” by “span”

Figure ED5: There are no blue lines in the Figure – just red and black lines.

140 – replace line by “and depleted dark haze particles on these slopes, thus revealing more of the water ice bedrock.

Supplementary Figure 1

How might one test this variable mixing ratio scenario?

Laurence Trafton

Reviewer #2 (Remarks to the Author):

Review of Nature Comm paper by Bertrand et al. on “How Methane Frost Forms on Mountain Tops and Crater Rims in Pluto's Equatorial Regions.”

As a glaciologist, I can say that this paper is one of the most intriguing ones I have read in quite a while. Overall, I find that it is well written and the methods are acceptable. In terms of methodology I have one major point to make and have struggled to find the right response, as I do not want to send this paper back to redevelopment, not do I want it to be rejected for this reason, but it is a very important area where at least future work can be improved. More about that later. Here is a point by point review of the paper's strengths and weaknesses or places where improvements should or could be made— most of them rather minor. I have made my notes or comments or suggested edits directly in the manuscript as an aid to this review. The manuscript notes are not supplemental to this written summary, but were just my running notes adapted into this summary. In all cases except those specifically noted in this review, there is nothing additional in the in-manuscript comments. The two major matters are the final two sets of comments.

Title: Right from the title onward, the paper is novel and exciting.

Abstract: It is clear and conveys the article's contents.

Page 3, 3 lines from bottom: Please describe very briefly what the Index Map is, what specifically it shows, and how it is different from a band-depth map.

Page 6, 3-4 lines from bottom. "In reality." An imagined reality. Maybe correct, but just starting with "The frost could have formed..." would be sufficient.

Page 6, last line. As a glaciologist, this paper is exciting, as mentioned. But that drives me to wonder whether it is possible that net Pluto-annual positive mass balances are possible and development of geologically or glaciologically significant masses are possible. Surely this is a question the authors have had. If there is a logical reason or model-based or back-of-envelope reason why this cannot be the case, then make that argument and dispel any notions of thick deposits. If you don't have an answer, or are saving it for another paper, at least ask the question somewhere in the paper, maybe in the Conclusions, and give the reader some reason to believe that this inevitable question is a shared perspective on the greater significance of this work.

Page 7 middle. "We determine that..." Is this purely an inference of the model? Is the basis a two-fold model inference: That there is the circulation pattern mentioned in the next two paragraphs, and also, in order to get condensation on the mountain tops there must be this enrichment in the CH₄:N₂ mixing ratio? So you have a need for that mixing ratio structure, and you have a mechanism to make it. So it's an inference based on those two points. An evident need for a mechanism, and the finding of a plausible mechanism... But is there any Pluto data from New Horizons bearing in the vertical CH₄:N₂ mixing ratio in the lower few kilometers of the atmosphere?

Page 8, nine lines from top: Is this meridional circulation and sublimation/condensation-induced wind pattern sustained on geological time scales, or is it an annually/seasonally oscillating pattern? To me it seems likely to be a seasonal variation, oscillating on the Pluto-annual time scale. Will there be a net annual positive mass balance anywhere? The simulation used as a basis was 30 million years long, and just the last few Earth decades of the data were used for this paper. I understand that this was to understand the recent snowy or frosty mountain weather. So that's fine. But I arrive back at two comments above this one. Somewhere the broader relevance in terms of possible glaciologically or geological deposits should be addressed, at least at the level of a question without an answer, or else as a logical argument that discounts the possibility of thick deposits. I have a hunch that another exciting paper is in the offing.

Page 8, eight lines up from the bottom. I suggest deleting the quotes around plutonian. That is a perfectly fine adjective; no need to seek pardon for its use via quotes.

Page 15, 6 lines from bottom, about 0.3% CO. Here I just make a note that the just-mentioned paper by Tan & Kargel used 0.01% CO. Maybe we were wrong. So I am just noting the difference, which very possibly is our shortcoming. We referenced our lower number for CO from Lellouchi et al. 2011.

I have no comments or suggestions regarding the supplement, which is a very nice addition to the paper.

MAJOR POINTS

Figure 1: This is a fascinating figure. I have a number of minor and some significant suggestions for Figure 1. (i) Add band depth scale bars for panels D, E, and F. Please refer to the annotated manuscript. (ii) Add a distance scale bar in panel A. Please add it carefully. (iii) Please double check the scale bar in B. I believe it is a little bit wrong. The length of 100 km scale bar is a little bit too long it seems. Capitalize Swiss Alps. Also, I am unsure of the British English (for NATURE) spelling of Alps, but check that. (iv) Panel D is around 200 km long by my calculation. But why should I have to calculate it? Please provide an accurate scale bar. (v) Indications of a transition in CH₄:N₂ is fascinating. We expect solid solution behavior. But would this be more likely to represent differences in a patchwork of N₂-rich and CH₄-rich phases? Or is it an artifact of data resolution? (vi) For panel F caption, is it meant instead "(F) Same as D for the N₂ band depth map." (vii) If this modeling/hypothesis is correct, then it is slope, slope aspect, latitude (those three determining solar insolation and surface temperature) and elevation that counts. It would seem likely that this has been investigated as a part of the modeling. If so, it would be worth showing a more details about this. Possibly another panel showing the modeled surface temperature or the amount of supersaturation (that would include the information that determines surface temperature and the elevation.— all of the variables).

Page 15, 7-8 lines from bottom, sentence that now reads: "In this context and awaiting more sophisticated schemes for simulating the behaviour of the different ice mixtures, we use Raoult's law as a simplification of the ternary phase diagram."

Such a 'more sophisticated scheme' is already published and has been applied to Pluto's solid-solid-vapor equilibria. See Tan and Kargel 2018. Tan, S.P., and J. S Kargel, 2017, Solid-phase equilibria on Pluto's surface, *Monthly Notices of the Royal Astronomical Society*, 474 (3), 4254–4263, <https://doi.org/10.1093/mnras/stx3036>.

For the editor's and authors' convenience, I attach a copy of it. Our model, CRYOCHEM, has been well validated and considers nonideality, which is significant in the N₂-CH₄-CO system. However, the qualitative storyline developed through this manuscript's (Bertrand et al.'s) modeling and the observations appear unlikely to be impacted strongly. The quantitative details would shift. For future work, I recommend adopting CRYOCHEM, which is a better EOS by hydrocarbon-rich mixtures (it also includes CO and N₂ in addition to CH₄ and others). The authors, of course, are free to dispute the advances that Tan and I made. The more pertinent matter is that is the Tan & Kargel paper might fit the calls be Bertrand et al. for an improved thermodynamic approach, which in my view we provide. Our applications have mainly been to Titan's liquid-liquid-vapor equilibria at its surface and to the tropopause, so this published paper I refer to is the only one so far that tackles Pluto's solid surface-vapor equilibria and that low a range of temperatures. There are still some big gaps in CRYOCHEM's development, but for Pluto's surface and the N₂-CH₄-CO system it is probably an adequate and substantial improvement over Raoult's "Law." I am not asking the authors to go back and rework their calculations. (Of course they could decide to do so. I am not asking for tha, as I'd like to see timely publication of this paper.) I have considered it qualitatively and suspect that nothing would change qualitatively. But the sentence at issue (from page 15) should be modified at the very least. Maybe replace it with this; the authors may decide on a different language:

"We note that sophisticated equations of state exist for the system N₂-CH₄-CO under Pluto surface conditions and have been applied to Pluto's surface ices and atmospheric vapor, though not in a GCM (CRYOCHEM, Tan and Kargel 2018). However, for simplicity in coding with a GCM, we have adopted Raoult's Law. As Tan and Kargel (2018) have shown with CRYOCHEM, this chemical system does not exhibit ideal behavior, but the approximation using Raoult's Law is sufficient for our present needs."

We thank the reviewers for their careful reading of our manuscript and the insightful comments. Below we provide point-by-point responses to the comments of the reviewers.

Note that:

- We improved a small detail of our reference simulation. In the previous simulation, paleo-deposits of CH₄ ice and subsequent albedo/emissivity changes led to slightly higher surface/soil temperatures within the band 30°S-20°S than what would have been produced if these terrains remained volatile-free at all times. This explains why there were no CH₄ ice accumulating within the band 30°S-20°S on the previous Fig 2. The new simulation includes a consistent decrease of surface temperature from the southern to the northern region of Cthulhu in 2015, see Supp Figure 1.A-C). As a result, CH₄ frosts consistently form on top of the entire mountain chains Pigafetta and Elcano Montes (Figure 2), in good agreement with New Horizons observations.
- We added a subpanel to Figure 3 showing surface temperatures.
- We added GCM maps of surface temperatures, CH₄ mass mixing ratio at saturation and net daily deposition rates in the supplementary materials.

Our comments are in red and added text is in blue.

REVIEWER COMMENTS

Reviewer #1 (Remarks to the Author):

This paper reports the most recent results of an ongoing line of investigation into processes affecting Pluto's seasonal evolution based on observations obtained during the 2015 flyby by New Horizons and a Pluto global circulation model developed and periodically extended by the leading authors since then. The focus of the present paper is on the interpretation of Pluto's methane-rich surface ice deposits, including their origin on mountain tops and crater walls, ridges and crests, and the formation of the bladed terrain. Processes forming these deposits are investigated using numerical simulations of Pluto's climate that include their existing model of the methane cycle. Based on these simulations, a process is proposed for forming the high-altitude deposits that is quite different from the adiabatic cooling of air uplifted by terrestrial mountains. Instead, mountain top/ridge methane condensation would be favored by sublimation-induced circulation cells that super-seasonally enrich atmospheric methane at these higher altitudes.

Pluto science has a high level of visibility and community interest owing to the rich store of data returned by that mission that is still being analyzed. These results will therefore be of wide interest to a multidisciplinary science community. Also included in this paper and the Supplementary Discussion are several highly-enhanced data products from the mission, with color stretched for clarity, showing unprecedented detail in the topographic distribution of methane and a red material. These figures will be of special interest to the readership. In particular, they support the modeling and the identification of the mountain top deposits as methane-rich ice. The authors suggest the red material may be ancient, highly processed, methane that has been exposed over super seasons.

However, the interpretation - or perhaps wording - in the text sometimes confuses a successful simulation with reality. Sometimes a simulated process is treated as a candidate or proposed hypotheses, and the result is properly evaluated in terms of consistency with the observations, with a good fit leading to the basis for making a prediction for further testing; e.g., lines 130-160 in the MS Word document. Otherwise a simulation result in agreement with observation is treated as a successful test of an included hypothesized process; or a process discovered in a simulation that successfully fits the observations is sometimes held up to be actually occurring on Pluto, rather than being a promising candidate. Such is the case for the process proposed for depositing frost at high altitudes. The Abstract asserts that processes on Pluto forming the high-altitude deposits are completely different than on Earth, do not result adiabatically, but result from a circulation cell. In verifiable reality, these are predictions of the model, as confirmed on line 73. This process may well be correct. But these simulations do not prove that. No evidence is given that these simulations uniquely represent the observational data set. Another model with a different driving hypothesis might equally fit the data. Results from simulations are model-dependent and their validity depends on the completeness and uniqueness of the model.

We agree that we have to be careful in the interpretation/wording of the model results. We acknowledge that our interpretation of the high-altitude CH₄ deposits may not represent a unique solution (although we are confident that the processes we describe play the major role, see below). Considering this, we reviewed the wording throughout the entire paper to address the concerns of this comment.

We want to emphasize that the model we developed is a full, fairly comprehensive, realistic model of Pluto's atmosphere simulating the 3D processes at work. Here we show that it simulates the formation of CH₄ deposits at high altitude as observed in a spontaneous and self-consistent way, without any ad hoc tuning.

There is also a certain degree of completeness and uniqueness of the model results, which are not sensitive to the free parameters of the model.

(1) The general circulation (and the near surface winds and transport of gaseous CH₄ and CO) is not sensitive to the ice distribution outside Sputnik Planitia. This was shown in the simulations by Bertrand et al., 2020 with the same GCM (attached, see Fig. 5). A global "retro-rotation" is forced by the conservation of angular momentum of air particles flowing above the equator in the condensation-sublimation flow. The model indicates that the circulation across the Sputnik Planitia basin is sufficient to force the entire zonal circulation.

(2) In our simulations, the ice distribution is in agreement with New Horizons observations. CH₄-rich ice deposits cover the northern mid-to-polar latitudes and the locations of the Bladed Terrain Deposits, and N₂-rich ice covers Sputnik Planitia and mid latitudes, as in Bertrand et al. (2020). The model then self-consistently and spontaneously predicts the CH₄-enriched layer at ~4 km altitude in the equatorial regions, as a result of the North to South circulation and vertical upward motions induced by the topography of Sputnik Planitia. In fact, in the first GCM simulations of Pluto, Forget et al. (2017) also found that the atmosphere was, to some degree, enriched in CH₄

at high altitude above the equatorial regions (see their Fig. 12) as CH₄ was transported from the northern to the southern hemisphere in their simulations.

We obtain qualitatively similar results if the Bladed Terrains Deposits are removed, or if the N₂ condensation-sublimation cycle or the diurnal cycle are turned off. CH₄ -rich frost always forms in larger amounts on top of the mountains but then the details of the net diurnal budget depend on the sources (CH₄ sublimation rates over the Bladed Terrains Deposits and the polar deposits) and sinks (condensation rates on top of the mountains and slopes, albedo feedbacks).

A particular pathway through which the employed model may prove to be non-unique, or even wanting, is the simplification (given in the Methods section) of using Raoult's law to treat condensation-sublimation of the CH₄:N₂:CO solid solution, rather than using the more complicated ternary phase diagram. This would require knowledge of the equation of state that is not generally available, and such theoretical ternary diagrams are not yet experimentally verified.

We thank the reviewer for discussing this point. Taking into account a ternary phase diagram in the model would introduce the hypothesis that the volatile ices relax toward an equilibrium state (over a specific depth and with different timescales that remain unknown), which would affect the ice sublimation and condensation rates by few percents (see below).

Considering this, we added in the Methods a few sentences to acknowledge the existence of the sophisticated equations of state for the N₂-CH₄-CO system. We also state that using Raoult's law gives some uncertainties of few percent on the CH₄ sublimation and condensation rates.

We also added in the text the following sentence:

The lack of image resolution, the uncertainties on the starting season of the frosts and the model approximations for computing CH₄ condensation and sublimation rates (see *Methods*) prevent us for estimating the thickness of CH₄-rich deposits with great confidence. Nevertheless, the 2015 simulation sheds light on the possible processes forming the CH₄-rich deposits seen at mountain summits in eastern Cthulhu.

For our paper, we feel that using Raoult's law remains a simplification that is legitimate, for the following reasons:

(1) The fact that Pluto's CH₄ ice is not at equilibrium and that its mixing ratio on the surface and in the atmosphere is rather dominated by dynamical processes (condensation, sublimation, atmospheric transport) than by the ternary phase equilibrium (which may also play a role, but the relaxation timescale may just be too long compared to that of the other processes).

Several observations support this point:

- a) The atmospheric mixing ratio of CH₄ is ~0.3% (Young et al. 1997; Young et al. 2018), 10 to 100 times higher than that expected by the 3-phase equilibrium
- b) The dilution of CH₄ in the N₂-rich terrains is only ~0.3 to 0.5% (Protopapa et al. 2017; Douté et al. 1999), rather than the ~4% expected for 3-phase equilibrium.
- c) CH₄-rich terrains dominate the surface northward of 55°N, with significant CH₄-rich terrain at low northern latitudes (20°N to 35°N) and in the massive, low latitude Bladed Terrain Deposits (Protopapa et al. 2017; Schmitt et al. 2019), and, at high altitude on Pigafetta and Elcano Montes (which is the focus of this paper).

- d) The fraction of CH₄-rich ice is slightly larger in northern Sputnik Planitia, where the dilution of CH₄ in N₂-rich ice is also larger (Protopapa et al. 2017; Schmitt et al. 2019).

We interpret Pluto to be a non-equilibrium dynamical environment with continuous exchange of materials. In this context, using the ternary phase diagram (which we agree is not easily available nor yet experimentally validated) as the only process to drive the partial pressure and ice mixing ratio would be wrong, in our opinion.

(2) The fact that we lack information to take into account the details of the dynamical processes involved with volatile ice sublimation-condensation (e.g. stratification, timescales for relaxation of the ices toward equilibrium, N₂ sublimation below a CH₄-rich layer through cracks, etc.)

Here we give some more details on N₂-CH₄ interactions but we want to point out that they should not concern the CH₄-rich deposits on top of Pigafetta Montes (the focus of this paper) since we believe that these deposits did not form from an initial N₂ ice, nor contain significant amounts of N₂ ice (as demonstrated by our Figure 1). We just want to emphasize how the ice mixtures on Pluto's surface may be dominated by the volatile ice sublimation-condensation rather than by the ternary phase diagram.

Laboratory experiments showed that processes of stratification (CH₄-rich ice on top of N₂-rich ice) are likely to widely occur on Pluto. Stratification processes could also reconcile the different spectral observations of Pluto's N₂ and CH₄ ices.

<https://www.hou.usra.edu/meetings/plutosystem2019/pdf/7004.pdf>

<https://www.abstractsonline.com/pp8/#!/9052/presentation/935>

On a flat N₂-rich surface, temperatures remain relatively constant but large amounts of N₂ ice are involved in condensation/sublimation, which could create a CH₄-rich ice layer on top. This may be what is happening to the northern edge of Sputnik Planitia, where N₂ constantly sublimates (polar day) and where CH₄-rich ice is detected in larger amounts and has a higher mixing ratio into N₂-rich ice than elsewhere in Sputnik Planitia.

To support this point, here is a quick calculation:

The phase diagram indicates that when N₂ sublimates, a CH₄-rich phase appears. The sublimation rate of CH₄ in this phase is about 8000 times lower than the N₂ sublimation rate (according to the phase diagram) while the sublimation rate of CH₄ in the N₂-rich phase is only 200-500 times lower than the N₂ sublimation rate (with 0.2-0.5% CH₄). Consequently, the CH₄-rich layer grows 15-40 times faster than it sublimates.

However, it is not clear how the surface temperature of such terrains evolves. Most likely, the CH₄-rich layer should remain at the N₂ ice temperature for a while and grow further. At some point the CH₄-rich layer would grow until a critical size for which it « isolates » the N₂-rich ice underneath (from absorption / emission / gas fluxes), which, without energy supply, should slow down its sublimation.

The issue is that the mechanisms of formation of such a CH₄-rich layer remain largely unknown and are therefore difficult to implement in the model.

In our model, we neglect any transient phase, and assume that the ice on the surface behaves as a N₂-rich or CH₄-rich deposit. In a way, this is equivalent to assuming that the timescale for relaxation toward equilibrium in the solid is much longer than the timescale of the sublimation-condensation dynamical processes.

In the future, we plan to investigate further all these mechanisms (stratification and relaxation toward equilibrium), develop new methods to implement them in the volatile transport model in a consistent way, and test them on Pluto and Triton. Such a work would require close collaboration and synergy between models (volatile transport, ternary phase equilibrium e.g. CRYOCHEM) and laboratory experiments.

(3) The fact that here we focus on CH₄-rich deposits only, that form from atmospheric CH₄ condensation on an initially volatile-free surface

In this paper, we hypothesized that the surface of Cthulhu is initially volatile-free (at least during part of the year) but becomes sufficiently cold at night (~42 K) to trigger CH₄ condensation. Therefore, we are not considering the case of a N₂-rich deposit that would accumulate CH₄ and become CH₄-rich. Instead, we are considering the case of CH₄-rich deposits that directly form on the surface with little fraction of N₂. The mixture cannot easily deviate toward a binary mixture CH₄-rich:N₂-rich at equilibrium considering the large atmospheric mixing ratio of CH₄, 100 times higher than expected by the 3-phase equilibrium. This is what LEISA observed (the deposits are CH₄-rich).

In the model, we make the assumption that the CH₄-rich deposits that would form in such conditions behave like pure CH₄ deposits, in terms of temperature and vapor pressure at saturation of CH₄. This is supported by our Figure 1.

If we assume that a CH₄ deposit is in equilibrium with the atmosphere, then the equilibrium vapor phase must be close to what is given in Table 2 in Tan and Kargel for 41.5 K, with 0.42% N₂ in CH₄-rich:N₂ ice. The partial pressure of CH₄ would be 0.00336 in this case, versus 0.00334 for pure CH₄, which is an error of less than 1%.

Considering that there is 0.3% of CH₄ in Pluto's atmosphere in 2015, sublimation would occur for pure CH₄ at 42.1 K. If the CH₄-rich ice were saturated with N₂, the critical temperature for CH₄ sublimation would be 42.4 K, only 0.3 K warmer than if the CH₄-rich ice were pure. Considering the uncertainties in albedo and emissivity, this is a small difference. During daytime, when the CH₄ ice surface is heated by the Sun and gets warmer, the surface temperature and thus the atmospheric mixing ratio at saturation q_0 strongly increase, and $q_0 \gg q_z$ (CH₄ mixing ratio just above the surface), leading to CH₄ sublimation for all pure CH₄ or CH₄-rich terrains with negligible differences.

Now, if we assume that the ice mixture relaxes towards the saturated 3-phase system (although we do not think it is the case), then at 42.0 K, the ratio of CH₄ ice's partial pressure to its pure vapor pressure is 78%. At higher daytime temperature, it would be higher and closer to 100% so the 22% approximation in this specific case is an upper limit.

We also note that the lack of experimental data and approximations may lead to some error of a few % in the available phase diagrams and EOS, which is comparable to the errors of our assumption.

As noted by Reviewer 2, our results remain qualitatively correct, and could be slightly shifted quantitatively if ice mixtures on Pluto actually evolve fast toward 3-phase equilibrium (although this is not strongly supported by the observations).

Also, regarding the issue of uniqueness, Tan & Kargel (2018) have previously proposed a competing process for explaining the deposition of the methane-rich solid solution on mountain tops/ridges that is based on their theoretical model of the equation of state. They show that local deviations from thermal equilibrium can sensitively result in the formation of methane-rich or nitrogen-rich ice according to the circumstance of the deviation. Their Fig. 2b, for Pluto, indicates the transformation of the nitrogen-rich phase VS_1 at low methane mole fractions to the methane rich phase VS_2 at ambient isobaric temperatures higher than for phase equilibrium. This would occur at high elevations, where the atmosphere is warmer than the isobaric condensation temperature.

We agree on the general idea that N₂-rich or N₂-rich:CH₄-rich ices may relax toward a CH₄-rich phase if there is an excess of CH₄ in the atmosphere or if the surface temperature increases locally.

However, here in the case of the Pigafetta Montes, the scenario suggested by the reviewer seems to assume that N₂-rich ice deposits initially form on top of the mountains, which is very unlikely considering the low albedo of Cthulhu and the fact that N₂ preferentially condense at low altitude for fundamental reasons (atmosphere-topographic process described by Trafton et al. 1998, Stansberry et al. 2014, and Bertrand and Forget 2016). Second, it seems to assume that the N₂ ice has a different temperature on top of the mountain than in the valley, but under the same atmospheric pressure, which cannot be the case on Pluto. It may be the case in a static system, without exchange / condensation / sublimation, but this remains very theoretical. Consequently, we do not think that this is a viable mechanism for the process described in this paper.

We also note that Tan and Kargel 2018 did not really explicitly (nor implicitly) mentioned this as a possible process for explaining the presence of CH₄-rich deposits at high altitude. Is this theory published elsewhere?

We calculate that on Pluto the ices are in a state of disequilibrium mostly driven by diurnal variation of insolation, atmospheric composition, surface temperatures, sublimation-condensation, segregation etc. rather than by the 3-phase equilibrium. The dynamic processes have different timescales (and involve significant ice thicknesses), and we think that they would be much shorter than the relaxation timescale within the solid. This calls for more laboratory experiments to test the different scenarios.

Although Tan & Kargel have been referenced (on lines 334-335), the reference is only in the context of the ternary diagram, not in terms of their proposed explanation for the deposition on

mountain ridges (among other Pluto ice phase phenomena).

We did not find this proposed explanation in Tan & Kargel, 2018, therefore this comment is difficult to address.

I do not recommend publication until these issues have been addressed satisfactorily. However, I believe there is enough new science and potential interest in this paper that a revised version should then merit publication in Nature. At a minimum, this paper should acknowledge and compare to the previous work, and place the proposed cellular deposition process in the context of an alternative model. The revised text and affected results of the fits should frame the conclusions as conditional, as has properly been done elsewhere in the text. This would be facilitated by making the suggested substitutions below.

Minor comments/corrections:

Define “phase-index” (as in maps).

The definition of phase index (or spectral index) is now given in Methods.

The spectral index, as defined in Schmitt et al. (2017) is an index allowing discrimination between CH₄ diluted in nitrogen ice and CH₄-rich ice phases based on the position of a set of CH₄ near-infrared bands. It is thus based on the measure of the shift of these bands upon dilution in nitrogen ice as measured in the laboratory (Quirico and Schmitt, 1997).

Line 114 – “CH₄ gas”? Yes, corrected

115 – “..., which suppresses rapid methane sublimation.” We reformulated this sentence: “except when nitrogen ice is present on the surface, because local N₂ ice sublimation can fill the first kilometres above the surface with cold air^{7,13}”

115 – “...local radiative balance everywhere.” Yes, corrected

119 – replace “show” by “indicate” or “suggest”. Corrected

120- replace “it is” by “It would be”. Corrected

121- replace “must” by “would have to be”. Corrected

166 - replace “shows” by “indicates” or “suggests”. Corrected

172 – ...mixing ratio in the model... Yes, corrected

176 – replace “We also found...” by “The model also predicts...” Corrected. We used it for the next sentence, which is more accurate.

185 – replace “shows” by “indicates” Corrected

194, - put comma after “Overall” Done

196 – replace “The “plutonian” process...” by “The proposed “plutonian” process...” Done

200 – replace “will” by “may” Done

201 - replace “will” by “may” Done

Figures

262 – replace “Same as A...” by “Same as D...” Done

265 – replace “than” by “as” Done

349 – append to sentence: “..., as the temperature is lowered.” Done

References on page 15 do not continue the numbering of page 9.
List “20 to 25” instead of “18, 19, 20, 21, 22, 24” **Corrected**

Supplementary Discussion
Line 47 – Define “spectral index”

The definition of phase index (or spectral index) is now given in Methods.
The phase index, as defined in Schmitt et al. (2017) is an index allowing to discriminate between CH₄ diluted in nitrogen ice and CH₄-rich ice phases based on the position of a set of CH₄ near-infrared bands. It is thus based on the measure of the shift of these bands upon dilution in nitrogen ice as measured in the laboratory (Quirico and Schmitt, 1997).

60 – replace “because of “ by “which we attribute to” or by “which the modeling attributes to”
Corrected

75 – replace “receive” by “received” **Corrected**

103 – replace “indicate” by “span” **Corrected**

Figure ED5: There are no blue lines in the Figure – just red and black lines.

Figure ED5 does have a blue line in the word and in the pdf document that I submitted. I am not sure I understand what happened to this figure.

140 – replace line by “and depleted dark haze particles on these slopes, thus revealing more of the water ice bedrock.

Corrected

Supplementary Figure 1

How might one test this variable mixing ratio scenario?

We want to point out that this is not a “scenario” that we prescribe but a robust outcome of Global Climate Modelling that is the topic of our article. Of course it would be wonderful to obtain observations that could confirm these simulations. However this is difficult to observe... New Horizons retrieved a CH₄ mixing ratio down to 80 km using the Alice instrument (UV spectrometer), but not below because the increasing CH₄ gas density saturated the signal. In addition, the UV measurements were less sensitive to CH₄ at low altitude because most of the photons at wavelength < 140 nm are absorbed at higher altitude.

NIR or Mid-IR observations of Pluto’s reflected light with high spatial resolution could allow for CH₄ gas detection (in absorption, the entire column would be measured). Locally, we could detect regions where the atmosphere is enriched in CH₄ gas, and maybe see an effect associated with these CH₄ layers. It could be seen by JWST/NIRSpec, for instance.

Laurence Trafton

Reviewer #2 (Remarks to the Author):

Review of Nature Comm paper by Bertrand et al. on “How Methane Frost Forms on Mountain Tops and Crater Rims in Pluto's Equatorial Regions.”

As a glaciologist, I can say that this paper is one of the most intriguing ones I have read in quite a while. Overall, I find that it is well written and the methods are acceptable. In terms of methodology I have one major point to make and have struggled to find the right response, as I do not want to send this paper back to redevelopment, not do I want it to be rejected for this reason, but it is a very important area where at least future work can be improved. More about that later. Here is a point by point review of the paper's strengths and weaknesses or places where improvements should or could be made— most of them rather minor. I have made my notes or comments or suggested edits directly in the manuscript as an aid to this review. The manuscript notes are not supplemental to this written summary, but were just my running notes adapted into this summary. In all cases except those specifically noted in this review, there is nothing additional in the in-manuscript comments. The two major matters are the final two sets of comments.

Title: Right from the title onward, the paper is novel and exciting.

Abstract: It is clear and conveys the article's contents.

Page 3, 3 lines from bottom: Please describe very briefly what the Index Map is, what specifically it shows, and how it is different from a band-depth map.

The definition of phase index (or spectral index) is now given in Methods.

The phase index, as defined in Schmitt et al. (2017) is an index allowing discrimination between CH₄ diluted in nitrogen ice and CH₄-rich ice phases based on the position of a set of CH₄ near-infrared bands. It is thus based on the measure of the shift of these bands upon dilution in nitrogen ice as measured in the laboratory (Quirico and Schmitt, 1997).

Page 6, 3-4 lines from bottom. “In reality.” An imagined reality. Maybe correct, but just starting with “The frost could have formed...” would be sufficient.

Corrected.

Page 6, last line. As a glaciologist, this paper is exciting, as mentioned. But that drives me to wonder whether it is possible that net Pluto-annual positive mass balances are possible and development of geologically or glaciologically significant masses are possible. Surely this is a question the authors have had. If there is a logical reason or model-based or back-of-envelope reason why this cannot be the case, then make that argument and dispel any notions of thick deposits. If you don't have an answer, or are saving it for another paper, at least ask the question somewhere in the paper, maybe in the Conclusions, and give the reader some reason to believe that this inevitable question is a shared perspective on the greater significance of this work.

We regrouped this comment with the following one, see answer below.

Page 8, nine lines from top: Is this meridional circulation and sublimation/condensation-induced wind pattern sustained on geological time scales, or is it an annually/seasonally oscillating pattern? To me it seems likely to be a seasonal variation, oscillating on the Pluto-annual time scale. Will there be a net annual positive mass balance anywhere? The simulation used as a basis was 30 million years long, and just the last few Earth decades of the data were used for this paper. I understand that this was to understand the recent snowy or frosty mountain weather. So that's fine. But I arrive back at two comments above this one. Somewhere the broader relevance in terms of possible glaciologically or geological deposits should be addressed, at least at the level of a question without an answer, or else as a logical argument that discounts the possibility of thick deposits. I have a hunch that another exciting paper is in the offing.

This is a good point. We detail the answer below (the first 2 paragraphs provide some context that seemed important to fully address the question).

First, regarding the general circulation regime of the atmosphere and its seasonal variation, our GCM results suggest that the retro-rotation regime is maintained during most of Pluto's year, with a maximum westward wind of $\sim 10\text{-}12\text{ m s}^{-1}$ centered above Sputnik Planitia during northern spring and summer (see Fig. 8 and 9 in Bertrand et al., 2020). This is because in our simulations, there is always enough cross-equatorial transport of gaseous N_2 in Sputnik Planitia (and outside), from north to south during northern spring and summer or south to north during the opposite season, to trigger westward winds. We have not looked at what would happen at past epochs when Pluto's obliquity was higher or lower; we can imagine that the cross-equatorial transport of gaseous N_2 could be maintained inside Sputnik Planitia during most of Pluto's year and therefore the retro-rotation too, but it is not certain.

Second, in this paper we hypothesize that the CH_4 -rich ice seen on top of Pigafetta mountains is seasonal, based on the fact that (1) similar frosts are seen on north-facing slopes of craters in the surrounding area but not of south-facing slopes; if the amounts of CH_4 deposited on these slopes are the same as on top of Pigafetta mountains, then the CH_4 frosts there may also be seasonal, and (2) brighter and redder brownish colours seem to surround Pigafetta mountains and may be indicative of a more extended CH_4 frost coverage in these regions in a previous season. However, we agree that the CH_4 -rich deposits seen there by New Horizons may also be thicker deposits and may subsist there over multiple Pluto years. This would be the case if the amounts of CH_4 involved during the "condensation season" are large enough to subsist during the "sublimation season". Albedo feedbacks could help build larger deposits there than mm-thick frosts.

Now, would it be possible for climatic patterns across different timescales to ever allow accumulation of the methane deposits onto these mountains to such thicknesses that they might be significant in terms of affecting the geology and geomorphology of the landscape on which they're emplaced?

It seems unlikely. We have now expanded this final section of the supplementary information to talk more about the contrast between the small-scale methane deposits in eastern Cthulhu and the large-scale bladed terrain deposits that form Tartarus Dorsa, and which cover much of the low

latitudes in the sub-Charon hemisphere. It has been noted before that there seems to be a surficial composition sequence within Pluto's equatorial uplands, from dominance by N₂ and CH₄ ice east of Sputnik, the abundance of which falls off to leave a mostly volatile-free landscape in east Cthulhu. This global-scale, longitudinal asymmetry in volatile abundance likely has an explanation that is rooted in climate dynamics, but investigating it is beyond the purview of this study. What does seem to be the case is that climatic conditions in east Cthulhu are such that condensation of CH₄ ice is limited to select areas including mountaintops and crater rims, and it cannot accumulate in large enough amounts to create significant geological consequences. Besides the possibility that the especially reddish areas around these deposits may indicate past, expanded coverage of the CH₄ ice, there seems to be no geomorphological evidence of expanded, thicker deposits. We note that there are mountain valleys surrounding Pigafetta Montes, and narrower, more incised valleys surrounding Elcano Montes, but N₂ ice is the more likely volatile to have carved any valleys through glacial flow (as described in the glacial study of Howard et al., 2017). CH₄ ice is relatively rigid and doesn't really flow at Pluto conditions (Moore et al., 2017a, Eluszkiewicz, and Stevenson, 1990. Rheology of solid methane and nitrogen – Applications of Triton, GRL).

Added text in the paper outside the supplementary materials:

It may also be possible that they grew thicker (up to a few meters) over multi-annual timescales, during past climate epochs with larger amounts of gaseous CH₄ available for condensation.

We also added this discussion in the Supplementary Materials

Page 7 middle. “We determine that...” Is this purely an inference of the model? Is the basis a two-fold model inference: That there is the circulation pattern mentioned in the next two paragraphs, and also, in order to get condensation on the mountain tops there must be this enrichment in the CH₄:N₂ mixing ratio? So you have a need for that mixing ratio structure, and you have a mechanism to make it. So it's an inference based on those two points. an evident need for a mechanism, and the finding of a plausible mechanism... But is there any Pluto data from New Horizons bearing in the vertical CH₄:N₂ mixing ratio in the lower few kilometers of the atmosphere?

No, the vertical CH₄ mixing ratio in the lower few kilometres of Pluto's atmosphere remain unobserved.

However, we want to emphasize that there is no tuning or forcing in our model aimed at producing a CH₄-enriched layer at ~4 km altitude. In our simulations, the ice distribution is in agreement with New Horizons observations. CH₄-rich ice deposits cover the northern mid-to-polar latitudes and the locations of the Bladed Terrain Deposits, and N₂-rich ice covers Sputnik Planitia and the mid latitudes, as in Bertrand et al. (2020). The model then self-consistently and spontaneously predicts the CH₄-enriched layer at ~4 km altitude in the equatorial regions, as a result of the north to south circulation and vertical upward motions induced by the topography of Sputnik Planitia.

In fact, in the first GCM simulations of Pluto, Forget et al. (2017) also found that the atmosphere was, to some degree, enriched in CH₄ at high altitude above the equatorial regions (see their Fig. 12) as CH₄ was transported from the northern to the southern hemisphere.

Page 8, eight lines up from the bottom. I suggest deleting the quotes around plutonian. That is a perfectly fine adjective; no need to seek pardon for its use via quotes.

Corrected

Page 15, 6 lines from bottom, about 0.3% CO. Here I just make a note that the just-mentioned paper by Tan & Kargel used 0.01% CO. Maybe we were wrong. So I am just noting the difference, which very possibly is our shortcoming. We referenced our lower number for CO from Lellouch et al. 2011.

We use a CO ice mixing ratio into N₂-rich ice of 0.3%, as suggested by Merlin et al., 2015 (see their table 3).

<https://www.aanda.org/articles/aa/abs/2015/10/aa26721-15/aa26721-15.html>

Lellouch et al., 2011, actually states: “However a refined analysis of the near-IR spectra indicates that CO is present on Pluto’s surface with a mixing ratio of 0.08–0.2% relative to N₂ (Douté et al. 1999).”

I have no comments or suggestions regarding the supplement, which is a very nice addition to the paper.

MAJOR POINTS

Figure 1: This is a fascinating figure. I have a number of minor and some significant suggestions for Figure 1.

(i) Add band depth scale bars for panels D, E, and F. Please refer to the annotated manuscript.

Done

(ii) Add a distance scale bar in panel A. Please add it carefully.

The scale across this scene varies dramatically in simple cylindrical projection, so we have decided not to show a scale bar and stick with just showing latitude and longitude lines. We do show a scale bar for the detail in panel B.

(iii) Please double check the scale bar in B. I believe it is a little bit wrong. The length of 100 km scale bar is a little bit too long it seems. Capitalize Swiss Alps. Also, I am unsure of the British English (for NATURE) spelling of Alps, but check that.

The original panel B showed a detail of the raw P_Color2 observation, across which the scale varied dramatically, but this has now been replaced with a simple cylindrical reprojection, and the scale bar has been modified accordingly.

(iv) Panel D is around 200 km long by my calculation. But why should I have to calculate it? Please provide an accurate scale bar.

Done.

(v) Indications of a transition in CH₄:N₂ is fascinating. We expect solid solution behavior. But would this be more likely to represent differences in a patchwork of N₂-rich and CH₄-rich phases? Or is it an artifact of data resolution?

At the spatial resolution of the LEISA measurements (2.7 km in the high resolution strip used for Figure 1.D and 1.F, and 7 km used for the 1.E panel) we cannot discriminate between a patchwork distribution of N₂-rich and CH₄-rich ice at a scale below a few kilometres, with an intimate mixture of the crystals of both phases or a vertical stratification (but at sub-mm scale). But, taking into account the noise and detection level, the maps show that nitrogen may be only present in some very localized areas at low altitude. The phase index maps clearly point to a CH₄-rich dominant composition (as the only phase present there).

We added this paragraph in Methods (Phase index maps)

(vi) For panel F caption, is it meant instead “(F) Same as D for the N₂ band depth map.”

Yes, Corrected

(vii) If this modeling/hypothesis is correct, then it is slope, slope aspect, latitude (those three determining solar insolation and surface temperature) and elevation that counts. It would seem likely that this has been investigated as a part of the modeling. If so, it would be worth showing a more details about this. Possibly another panel showing the modeled surface temperature or the amount of supersaturation (that would include the information that determines surface temperature and the elevation.— all of the variables).

We addressed this comment by modifying Figure 3 B and adding an extra panel showing the surface temperatures. The figure now shows that the CH₄ frosts are much colder than the dark volatile-free terrains during daytime (lower maximal temperatures), because of their higher albedo, and slightly warmer during nighttime (higher minimal temperatures), because of their lower emissivity (0.8 for the ice, 1 for the dark materials). As CH₄ ice accumulates and as its albedo increases, daytime sublimation decreases which allows for an increase of daily net ice accumulation. We also added the maps corresponding to these cross-sections in the supplementary materials.

Page 15, 7-8 lines from bottom, sentence that now reads: "In this context and awaiting more sophisticated schemes for simulating the behaviour of the different ice mixtures, we use Raoult's law as a simplification of the ternary phase diagram."

Such a 'more sophisticated scheme' is already published and has been applied to Pluto's solid-solid-vapor equilibria. See Tan and Kargel 2018. Tan, S.P., and J. S Kargel, 2017, Solid-phase equilibria on Pluto's surface, Monthly Notices of the Royal Astronomical Society, [474 \(3\), 4254–4263](https://doi.org/10.1093/mnras/stx3036), <https://doi.org/10.1093/mnras/stx3036>.

For the editor's and authors' convenience, I attach a copy of it. Our model, CRYOCHEM, has been well validated and considers nonideality, which is significant in the N₂-CH₄-CO system. However, the qualitative storyline developed through this manuscript's (Bertrand et al.'s) modeling and the observations appear unlikely to be impacted strongly. The quantitative details would shift. For future work, I recommend adopting CRYOCHEM, which is a better EOS by hydrocarbon-rich mixtures (it also includes CO and N₂ in addition to CH₄ and others). The authors, of course, are free to dispute the advances that Tan and I made. The more pertinent matter is that is the Tan & Kargel paper might fit the calls be Bertrand et al. for an improved thermodynamic approach, which in my view we provide. Our applications have mainly been to Titan's liquid-liquid-vapor equilibria at its surface and to the tropopause, so this published paper I refer to is the only one so far that tackles Pluto's solid surface-vapor equilibria and that low a range of temperatures. There are still some big gaps in CRYOCHEM's development, but for Pluto's surface and the N₂-CH₄-CO system it is probably an adequate and substantial improvement over Raoult's "Law." I am not asking the authors to go back and rework their calculations. (Of course they could decide to do so. I am not asking for tha, as I'd like to see timely publication of this paper.) I have considered it qualitatively and suspect that nothing would change qualitatively. But the sentence at issue (from page 15) should be modified at the very least. Maybe replace it with this; the authors may decide on a different language: "We note that sophisticated equations of state exist for the system N₂-CH₄-CO under Pluto surface conditions and have been applied to Pluto's surface ices and atmospheric vapor, though not in a GCM (CRYOCHEM, Tan and Kargel 2018). However, for simplicity in coding with a GCM, we have adopted Raoult's Law. As Tan and Kargel (2018) have shown with CRYOCHEM, this chemical system does not exhibit ideal behavior, but the approximation using Raoult's Law is sufficient for our present needs."

We corrected the sentence as suggested.

We want to emphasize that we are very interested to test the equations of state for the system N₂-CH₄-CO in our models in the future. The main issue for now remains that it is quite complex to implement in the models and adds in many new free parameters (timescales for the relaxation toward equilibrium, depth over which the mixing occurs, and overall how the sublimation and condensation rates are impacted remains unclear).

REVIEWER COMMENTS

Reviewer #1 (Remarks to the Author):

This version of the paper is much improved as the advised changes have been satisfactorily made, with one important exception: It remains for the authors to acknowledge previous work in the scientific literature on the topic of this paper – how CH₄ frost forms on Pluto's elevated topography. The issue appears to be that a specific example for the elevated altitude case was not given in the recommended reference, Tan & Kargel (2018), which chooses for examples the relatively warm volatile-free Cthulho Regio and relatively cold Tombaugh Regio. However, their Table 2 and Fig. 2b discuss results from the isobaric projection of the ternary diagram at Pluto's conditions during the NH flyby; and they cite the success of the isobaric limiting case in explaining Pluto's phenomena as a validation of the EOS model in general, where pressure and temperature vary together to include segregation by altitude. See also their Fig. 5, which shows the sublimation and condensation curves in the binary case in which temperature and pressure change together. There is only a very narrow gap in either dimension along the sublimation curve where N₂-rich ice changes to CH₄-rich ice (and vice versa).

As an aside, the reviewer did not make any assumption that N₂-rich deposits initially form on top of mountains in interpreting Fig. 2b. Instead, Fig. 2b shows how the phase equilibrium would change with a change in temperature in either direction; either rising from the N₂-rich ice phase to result in a CH₄-rich ice phase, or vice versa. The point is that the resulting phase is very sensitive to the direction of a departure from equilibrium. An added consequence would be that the error in using Raoult's law would be much bigger if the wrong phase were assumed due to such a switch have taken place. However, the authors are free to choose their own hypothesis, which does not have to be one that includes thermodynamic equilibrium. As they have pointed out, global thermal equilibrium does not strictly occur due to the high CH₄ mixing ratio.

In response to the authors' query about another publication that explicitly mentions segregation by altitude, I attach two further references by L. Young et al. - see the highlighted sentences. These are meeting abstracts submitted in 2019 to the Pluto System After New Horizons 2019 (LPI Contrib. No. 2133) and the EPSC-DPS Joint Meeting (EPSC Abstracts Vol. 13 EPSC-DPS2019-1015-1). As these abstracts are electronically searchable, they are part of the scientific literature and record. I note the first author of the submitted paper is a coauthor on both of these abstracts; however, the highlighted sections of both papers refer back to the refereed Tan & Kargel (2018) paper. Since these papers report prior research on the topic of the submitted paper, albeit using a different approach, they should be referenced in the submitted paper. I would be prepared to recommend publication in Nature after proper acknowledgement of the prior work.

Laurence Trafton

Two attachments

Reviewer #2 (Remarks to the Author):

Thank you for taking the care to respond clearly to my review (and the other reviewer, too). I have only a few relatively minor further comments and suggestions (contained in the marked-up PDF, which should be read carefully to see the comments). I list here only a few of them, but there is nothing I found that is a "make or break" issue. The manuscript is in very good shape, and it's about as exciting a manuscript as I have read in a long time. It's the kind of thing that makes me want a new mission!

1. Title: I slightly prefer the "alternative" title.

2. Line 109 and also line 211, on Earth snowfall from within-column condensation is the main process by which snow-caps of mountains form. It is overwhelmingly not "frost," which is a sublimative veneer of fine ice crystals deposited directly on a solid surface. On Pluto it may well be (seems to be) frost. So this is a further distinction between Earth and Pluto, according to your modeling. If Pluto's atmosphere snows, it's probably just in the lower boundary layer of air that undergoes surface-driven cooling and katabatic flow.

3. Lines 196-197: It should be stated explicitly that this CH₄:N₂ mixing ratio profile is a model outcome of the GCM and also required by the condensation model, though it is not a forced contrived outcome. However, New Horizons did not produce observations of such a mixing ratio (or any mixing ratio). This is as mentioned in the authors' response to my earlier review. I think it is important to state this explicitly in the paper, even though it is implicit in what is already written.

4. Methods section on "CH₄ and CO condensation-sublimation on the surface": The revised verbage is fine. (And I also agree with a response to reviewers that there can be much synergy in bringing CRYOCHEM more directly to bear on New Horizons observations, and vice versa, and bringing different modelers together.)

--Jeffrey S. Kargel

Planetary Science Institute

Below we provide point-by-point responses to the comments of the reviewers.
Our comments are in red and added text is in blue.

REVIEWER COMMENTS

Reviewer #1 (Remarks to the Author):

This version of the paper is much improved as the advised changes have been satisfactorily made, with one important exception: It remains for the authors to acknowledge previous work in the scientific literature on the topic of this paper – how CH₄ frost forms on Pluto's elevated topography. The issue appears to be that a specific example for the elevated altitude case was not given in the recommended reference, Tan & Kargel (2018), which chooses for examples the relatively warm volatile-free Cthulho Regio and relatively cold Tombaugh Regio. However, their Table 2 and Fig. 2b discuss results from the isobaric projection of the ternary diagram at Pluto's conditions during the NH flyby; and they cite the success of the isobaric limiting case in explaining Pluto's phenomena as a validation of the EOS model in general, where pressure and temperature vary together to include segregation by altitude. See also their Fig. 5, which shows the sublimation and condensation curves in the binary case in which temperature and pressure change together. There is only a very narrow gap in either dimension along the sublimation curve where N₂-rich ice changes to CH₄-rich ice (and vice versa).

As an aside, the reviewer did not make any assumption that N₂-rich deposits initially form on top of mountains in interpreting Fig. 2b. Instead, Fig. 2b shows how the phase equilibrium would change with a change in temperature in either direction; either rising from the N₂-rich ice phase to result in a CH₄-rich ice phase, or vice versa. The point is that the resulting phase is very sensitive to the direction of a departure from equilibrium. An added consequence would be that the error in using Raoult's law would be much bigger if the wrong phase were assumed due to such a switch have taken place. However, the authors are free to choose their own hypothesis, which does not have to be one that includes thermodynamic equilibrium. As they have pointed out, global thermal equilibrium does not strictly occur due to the high CH₄ mixing ratio.

In response to the authors' query about another publication that explicitly mentions segregation by altitude, I attach two further references by L. Young et al. - see the highlighted sentences. These are meeting abstracts submitted in 2019 to the Pluto System After New Horizons 2019 (LPI Contrib. No. 2133) and the EPSC-DPS Joint Meeting (EPSC Abstracts Vol. 13 EPSC-DPS2019-1015-1). As these abstracts are electronically searchable, they are part of the scientific literature and record. I note the first author of the submitted paper is a coauthor on both of these abstracts; however, the highlighted sections of both papers refer back to the refereed Tan & Kargel (2018) paper. Since these papers report prior research on the topic of the submitted paper, albeit using a different approach, they should be referenced in the submitted paper. I would be prepared to recommend publication in Nature after proper acknowledgement of the prior work.

We thank the reviewer for clarifying this point and for suggesting to cite the conference abstracts. Altitude segregation has been mentioned in these abstracts; however we note that the exact mechanism remains to be explored (in particular with models and lab experiments).

Nevertheless, we now explicitly mention this work in the main text of the paper and hope that we satisfactorily addressed the reviewer's comment:

It has been suggested that the sublimation and condensation of volatile ices could drive the ices out of thermodynamic equilibrium and result in altitude segregation with N₂-rich ice dominating at low-elevations and CH₄-rich ice dominating at high elevations [Tan and Kargel, 2018, Young et al., 2019]. Here we explore an alternative scenario that involves an atmospheric process.

Laurence Trafton

Two attachments

Reviewer #2 (Remarks to the Author):

Thank you for taking the care to respond clearly to my review (and the other reviewer, too). I have only a few relatively minor further comments and suggestions (contained in the marked-up PDF, which should be read carefully to see the comments). I list here only a few of them, but there is nothing I found that is a "make or break" issue. The manuscript is in very good shape, and it's about as exciting a manuscript as I have read in a long time. It's the kind of thing that makes me want a new mission!

We thank the reviewer for his compliments and we are glad that he finds the subject of the paper as exciting as we do!

1. Title: I slightly prefer the "alternative" title.

We agree that the alternative title is punchier. "Earth-like" may be confusing though because it is only Earth-like by appearance. Maybe:

"Equatorial Mountain Chains on Pluto are Covered by Methane Frosts Resulting from a Unique Atmospheric Process"
is a better choice, if the editor agrees.

2. Line 109 and also line 211, on Earth snowfall from within-column condensation is the main process by which snow-caps of mountains form. It is overwhelmingly not "frost," which is a sublimative veneer of fine ice crystals deposited directly on a solid surface. On Pluto it may well be (seems to be) frost. So this is a further distinction between Earth and Pluto, according to your modeling. If Pluto's atmosphere snows, it's probably just in the lower boundary layer of air that undergoes surface-driven cooling and katabatic flow.

Comment in the PDF

A meteorological definition of frost may be: a deposit of small white ice crystals formed on the ground or other surfaces when the temperature falls below freezing and below the saturation vapor pressure of H₂O. Condensation within the atmospheric column and snowfall is the main process on Earth for making snowpack on Earth. Direct sublimative frost condensation directly on the surface of course happens but it contributes negligibly to snowpack mass balance and remote sensing spectroscopy of most Earth mountains. So this is may be your chance to clarify what will be seen, by the end of the reading of this paper, another meteorological distinction between Earth's snowy mountains and Pluto's frost accumulations of methane ice. If either Earth's snowpack or Pluto's methane frosts accumulate to great thicknesses, then glaciation may result. Though Pluto's glaciers may be extra weird, because nitrogen also can flow.

We agree that it does not snow on Pluto. In the paper, snow is now only mentioned in the context of Earth. As suggested, we added a note in the paper to clarify the snow vs frost difference.

Note that here frost is defined as ice crystals that form directly on a below-freezing surface via a phase change from gas in the atmosphere, whereas snow is defined as individual ice crystals that grow while suspended in the atmosphere and subsequently fall as precipitation onto the surface.

In the PDF:

The glacially structured appearance of the Pluto Mountains is remarkable, though of course the image resolution is still quite low to be sure that the right geomorphological analog is used. This is, however, one of the aspects that is so intriguing here. The others are the methane ice caps. and the meteorology. But let's say the glacial geomorphology analog is correct. If it is the methane ice that's responsible, then it would imply that the long-term (let's say millions of Earth years) process involves net annual positive mass balance of methane in these mountains. Of course the inferred glaciation could involve an older period of methane ice accumulation, or the active glacial agent might be nitrogen, instead.

We agree with this comment. In order to better investigate this geomorphology, it would be of interest to run the climate model with the CH₄ cycle over several years (in order to capture the net annual balance for CH₄) or even better over astronomical timescales (with Milankovich-like obliquity variations for Pluto). We leave this for a future work.

3. Lines 196-197: It should be stated explicitly that this CH₄:N₂ mixing ratio profile is a model outcome of the GCM and also required by the condensation model, though it is not a forced contrived outcome. However, New Horizons did not produce observations of such a mixing ratio (or any mixing ratio). This is as mentioned in the authors' response to my earlier review. I think it is important to state this explicitly in the paper, even though it is implicit in what is already written.

OK, we added it explicitly in the text:

This vertical distribution of gaseous CH₄ in the first kilometres above the surface formed self-consistent as an outcome of our GCM simulation but remains unconstrained by observations as the CH₄ mixing ratio was not observed by New Horizons below 80 km altitude.

4. Methods section on "CH₄ and CO condensation-sublimation on the surface": The revised

verbage is fine. (And I also agree with a response to reviewers that there can be much synergy in bringing CRYOCHEM more directly to bear on New Horizons observations, and vice versa, and bringing different modelers together.)

Yes, this would correspond to the next step toward a better understanding of Pluto's ices.

In the PDF:

Thus, it may be recognized that Pluto lacks an Earth-like troposphere. But neither is the near-surface air a classical stratosphere or exosphere, because this surface-driven heat transfer process does disturb the stratification and produces weather. The prevalence of katabatic winds in this otherwise stratified layer could be termed a stratokatabatosphere if you want to coin a new tongue-twisting term. (You would have to say it quickly three times, and if the term survives your tongue, then use it!) It would seem that this special atmospheric layer would extend about as high, globally, as the mountain relief extends. Or is this not right?

Yes, it could be even more complicated than that because N₂ sublimation also perturbs the thermal profile, but only in the depression where N₂ ice is available. On top of the mountains however, it is pretty close to a stratosphere lying directly on top of the mountains despite few turbulences associated with surface winds.

We also addressed the suggested corrections that were embedded in the PDF.

--Jeffrey S. Kargel
Planetary Science Institute

REVIEWER COMMENTS

Reviewer #1 (Remarks to the Author):

I am satisfied with the present revision and now recommend this paper for publication in Nature.

Laurence Trafton

Reviewer #2 (Remarks to the Author):

GENERAL COMMENTS:

This whole 8 pages of reviewer commentary supports changes called to be applied to one paragraph of the manuscript. Almost all of the manuscript overall is in acceptable shape, the figures are improved a lot and are really beautiful and compelling portrayals of the data and ideas. Some corrections absolutely must be made in the revised text in one really bad paragraph in lines 381-394. As I am giving a “pass” in the most challenging weakness (poorly applicable Raoult’s Law), what remains to be corrected is easy— but necessary. There are some general points I make here, and in the “DETAILS” below I go into great detail mainly for future use regarding the core weakness, which has a route to improvement, but it will take significant work and may be out of scope for this paper. The ease of implementation of these fixes does not indicate that the manuscript skates by without underscoring a very serious weakness, which is only more apparent the more I considered it in light of errors made in the manuscript revision. I am driving home this weakness here to leave no doubt about it, and so the authors may understand that the more subtle acknowledgement of this problem that I am illuminating for the published paper is a requirement, in my view, to see publication... it can be subtly stated, but it must not be omitted. How I phrase and dwell on the problem in this review is for clarity, completeness, and to encourage a new approach in the future. I do not call for this brusque language to be implemented in the paper. But I want to be clear here, to the authors and editor. First thing, Raoult’s Law was at first intended EXCLUSIVELY for vapor-liquid equilibria and ideal solutions only. There is a special exception, and that is where an involatile solid is dissolved in the liquid, such as NaCl in liquid H₂O— this has become a very commonly and mainly accurately used variation on Raoult’s Law and sometimes is confused for the original “law.” None of those general types of applicable systems are involved on Pluto, so Raoult’s Law does not apply by conventional usage. Raoult’s Law commonly is further adapted by applying certain types of corrections to account for minor non-idealities encountered in liquid solutions of volatile materials. The manuscript takes a damn-the-torpedoes approach and pushes through assuming applicability of Raoult’s Law anyway in circumstances that no thermodynamicist accepts. This inevitably will

introduce substantial errors. I honestly don't know how big those errors are apt to be, whether they may invalidate this paper's conclusions or be merely a quantitative tweak. I hope for the latter. Because I cannot easily gauge how big those errors are, I am erring on the side of deference to the authors and am giving a "pass" on this weakness, but some changes are needed. This "pass" may be warranted also by the uncertainties in the basic observations of atmospheric properties, which may be where the biggest uncertainties lie. Furthermore, the lab-measured thermodynamic properties in the systems containing CO are not entirely good enough for our needs, but N₂-CH₄ is now pretty well known... and it is not a Raoult's Law kind of system. The main reason why a "pass" is provided on this weakness is because of the other strengths of the paper, which "DETAILS" also summarize. I would like to see this paper go to publication, as it will spur good discussions and further research, and I am hopeful that the thermodynamic weakness explained here will be resolved in future work by the authors or others. The new errors introduced into the paper clearly indicate that the authors have some ways to go to come up to speed on the thermodynamics and phase equilibria in this system. The "DETAILS" should help the authors going into the future and also help them understand the faults of their reliance on Raoult's Law. Below some specific changes are called for, and then the "DETAILS" justifies those required changes.

SPECIFIC CHANGES THAT ARE NEEDED (all are in one paragraph... lines 381-394):

The phrasing in lines 381-384 includes a first sentence and a first phrase of the second sentence that are correct but a little awkward; what follows is unclear and part is incorrect. Whether 3 phases are present depends on where in the system in composition and temperature; also pressure, but let's say that is almost fixed near the surface at 11 microbars. The manuscript says: "On Pluto's surface, the volatile ices should form solid solutions whose phases follow a ternary phase diagram.^{13,24} Thermodynamics models indicate that the chemical system N₂-CH₄-CO does not exhibit ideal behaviour system and should relax to a 3-phase equilibrium in which the sublimation pressure of each species depends only on temperature and is independent of the CH₄ and CO mole fraction^{13,24}."

This can be rephrased more simply and accurately to: "On Pluto's surface, the volatile ices should form solid solutions whose phases follow ternary phase equilibria; they do not exhibit ideal behaviour^{13,24}."

The next sentence in lines 385-387, can be kept as is: "We note that advances have been made in developing equations of state for the N₂-CH₄-CO system under Pluto surface conditions (CRYOCHEM¹³) and have been applied to Pluto's surface ices and atmospheric vapor, though not in a GCM."

The manuscript from line 387 to 390 (two sentences) is completely erroneous, as the "DETAILS" below elaborate. The manuscript says, "At the temperature of the CH₄ deposits modelled in this paper (~45 K), the ternary diagram predicts that the CH₄-rich ice should contain ~4% N₂. However, the predictions of the 3-phase system at equilibrium do not correspond to what has been observed

by New Horizons, and typically the ratio of N₂ into CH₄-rich ice could be much less.” This refers to a ternary phase diagram in Tan & Kargel, 2017, and is Figure 1 below (with annotated additions by me). As “DETAILS” show clearly, at 45 K the methane-rich phase in the binary N₂-CH₄ system (Figures 1 and 2 below) is nearly pure methane; it does not contain 4% nitrogen. This is seen also in the ternary diagram at 41.5 K (Figure 1; it gets only purer as temperature increases). Furthermore, at 45 K, the ternary system does not include three phases; it only includes two phases— methane-rich (nearly pure CH₄) solid and N₂-rich vapor. This pair of sentences is completely wrong. It must be changed. I recommend this language: “At the temperature of the CH₄ deposits modelled in this paper (~45 K, prevailing at high altitudes), the ternary phase equilibria shown by Tan and Kargel predict that two phases coexist: a very nearly pure CH₄ solid (<<1% impurities of N₂ and CO in solid solution) and N₂-rich vapor. Both the ternary and the binary systems shown by Tan and Kargel correspond closely to what has been observed by New Horizons and is consistent with what we report here in Figure 1. At lower temperatures at and below 36.9K (cold conditions prevailing in the lower elevations) the occurrence of a nitrogen-rich phase, according to the phase equilibria provided by Tan and Kargel¹³ also is like what New Horizons revealed in places like Sputnik Planitia; the methane-rich solid at those cold temperatures can contain much more impurities of N₂ and CO (up to a maximum of 4.6 molar % N₂ in solid solution at 36.9K). The impure (solid solution) N₂-bearing methane-dominated phase can coexist with vapor and an N₂-dominated phase below 36.9K¹³.”

Lines 390-394 also make a gratuitous statement about uncertainties. The manuscript says: “Here, for simplicity in coding with a GCM, we have adopted Raoult’s Law, as in previous GCM studies^{7,16,17}. This approximation is sufficient for our present needs and for the study of CH₄-rich ice deposits on Pluto’s surface, which are the focus of this paper, but we acknowledge that it leads to some uncertainties of few percent in the computation of the CH₄ sublimation and condensation rates.”

Given the mistakes made in lines 387-390, it is clear that the authors have actually not evaluated what errors may have been introduced by a reliance on Raoult’s Law or how other/better equations of state could affect the model output. The errors are potentially impactful on the major qualitative conclusions of this paper, not just on a few percent error in deposition and sublimation rates. It may affect the stability and mapped distribution of volatiles and the need for a methane-richer atmosphere at higher elevations. My deference to the authors is because I also lack my own full analysis of Raoult’s Law vs. better thermodynamics, but I make clear that no other science field to my knowledge— certainly not Chemistry— allows an application of Raoult’s Law to volatile solids’ equilibria with vapor. This selection of statements in lines 390-394 must be rephrased. I suggest this:

“Here, for simplicity in coding with a GCM, we have adopted Raoult’s Law, as in previous GCM studies^{7,16,17}. This approximation seems sufficient for our present needs and for the study of CH₄-rich ice deposits on Pluto’s surface, which are the focus of this paper, but we acknowledge that it leads to some unevaluated uncertainties in CH₄ solid phase stability.”

--Jeffrey S. Kargel

We thank the reviewer for the insightful comments, the very precise and helpful explanations on the N_2 - CH_4 thermodynamics, as well as for the detailed figures. Pluto is the perfect place to study the thermodynamics of N_2 - CH_4 and we aim at implementing more precise thermodynamics in our models in the future. In fact, we are excited to move forward on this topic and pursue the study of Pluto's regional and global climate, exotic surface and atmosphere with state-of-the-art climate models that are coupled with CRYOCHEM equations.

Below we provide point-by-point responses to the comments of the reviewer. We added some text in the method section to take into account these comments and to better emphasize the need for more N_2 - CH_4 -CO thermodynamics in KBO's climate models in the future.

REVIEWER COMMENTS

Reviewer #2 (Remarks to the Author):

GENERAL COMMENTS:

This whole 8 pages of reviewer commentary supports changes called to be applied to one paragraph of the manuscript. Almost all of the manuscript overall is in acceptable shape, the figures are improved a lot and are really beautiful and compelling portrayals of the data and ideas. Some corrections absolutely must be made in the revised text in one really bad paragraph in lines 381-394. As I am giving a “pass” in the most challenging weakness (poorly applicable Raoult's Law), what remains to be corrected is easy— but necessary. There are some general points I make here, and in the “DETAILS” below I go into great detail mainly for future use regarding the core weakness, which has a route to improvement, but it will take significant work and may be out of scope for this paper. The ease of implementation of these fixes does not indicate that the manuscript skates by without underscoring a very serious weakness, which is only more apparent the more I considered it in light of errors made in the manuscript revision. I am driving home this weakness here to leave no doubt about it, and so the authors may understand that the more subtle acknowledgement of this problem that I am illuminating for the published paper is a requirement, in my view, to see publication... it can be subtly stated, but it must not be omitted. How I phrase and dwell on the problem in this review is for clarity, completeness, and to encourage a new approach in the future. I do not call for this brusque language to be implemented in the paper. But I want to be clear here, to the authors and editor.

First thing, Raoult's Law was at first intended EXCLUSIVELY for vapor-liquid equilibria and ideal solutions only. There is a special exception, and that is where an involatile solid is dissolved in the liquid, such as NaCl in liquid H₂O— this has become a very commonly and mainly accurately used variation on Raoult's Law and sometimes is confused for the original “law.” None of those general types of applicable systems are involved on Pluto, so Raoult's Law does not apply by conventional usage. Raoult's Law commonly is further adapted by applying certain types of corrections to account for minor non-idealities encountered in liquid solutions of volatile materials. The manuscript takes a damn-the-torpedoes approach and pushes through assuming applicability of Raoult's Law anyway in circumstances that no thermodynamicist accepts. This inevitably will introduce substantial errors. I honestly don't know how big those errors are apt to be, whether they may invalidate this paper's conclusions or be merely a

quantitative tweak. I hope for the latter. Because I cannot easily gauge how big those errors are, I am erring on the side of deference to the authors and am giving a “pass” on this weakness, but some changes are needed. This “pass” may be warranted also by the uncertainties in the basic observations of atmospheric properties, which may be where the biggest uncertainties lie. Furthermore, the lab-measured thermodynamic properties in the systems containing CO are not entirely good enough for our needs, but N₂-CH₄ is now pretty well known... and it is not a Raoult’s Law kind of system.

We added a few sentences in the method to address this comment in the paper:

However, we note that Raoult’s Law was at first intended for vapor-liquid equilibria and ideal solutions only, which do not include the N₂-CH₄-CO system observed on Pluto. Despite the fact that this approximation gives good results and allows for reproducing the atmospheric mixing ratios observed by New Horizons observations¹⁷, it may still introduce errors on sublimation and condensation rates of the different types of ice.

We note that Figure 2 in Trafton 2015 seems to indicate that the error made with the Raoult’s law approximation on the vapor pressures is not significant when we assume CH₄-rich ices or N₂-rich ices. The error may be significant if we assume a mixture of saturated phases. However in our paper we only focus on “pure” CH₄ ice so the error may be acceptable.

The main reason why a “pass” is provided on this weakness is because of the other strengths of the paper, which “DETAILS” also summarize. I would like to see this paper go to publication, as it will spur good discussions and further research, and I am hopeful that the thermodynamic weakness explained here will be resolved in future work by the authors or others. The new errors introduced into the paper clearly indicate that the authors have some ways to go to come up to speed on the thermodynamics and phase equilibria in this system. The “DETAILS” should help the authors going into the future and also help them understand the faults of their reliance on Raoult’s Law. Below some specific changes are called for, and then the “DETAILS” justifies those required changes.

SPECIFIC CHANGES THAT ARE NEEDED (all are in one paragraph... lines 381-394):
The phrasing in lines 381-384 includes a first sentence and a first phrase of the second sentence that are correct but a little awkward; what follows is unclear and part is incorrect. Whether 3 phases are present depends on where in the system in composition and temperature; also pressure, but let’s say that is almost fixed near the surface at 11 microbars. The manuscript says: “On Pluto’s surface, the volatile ices should form solid solutions whose phases follow a ternary phase diagram.^{13,24} Thermodynamics models indicate that the chemical system N₂-CH₄-CO does not exhibit ideal behaviour system and should relax to a 3-phase equilibrium in which the sublimation pressure of each species depends only on temperature and is independent of the CH₄ and CO mole fraction^{13,24}.” This can be rephrased more simply and accurately to: “On Pluto’s surface, the volatile ices should form solid solutions whose phases follow ternary phase equilibria; they do not exhibit ideal behaviour^{13,24}.”

OK we agree with this statement and corrected the sentence.

The next sentence in lines 385-387, can be kept as is: “We note that advances have been made in developing equations of state for the N₂-CH₄-CO system under Pluto surface conditions (CRYOCHEM13) and have been applied to Pluto’s surface ices and atmospheric vapor, though not in a GCM.”

OK we are keeping the sentence as is.

The manuscript from line 387 to 390 (two sentences) is completely erroneous, as the “DETAILS” below elaborate. The manuscript says, “At the temperature of the CH₄ deposits modelled in this paper (~45 K), the ternary diagram predicts that the CH₄-rich ice should contain ~4% N₂. However, the predictions of the 3-phase system at equilibrium do not correspond to what has been observed by New Horizons, and typically the ratio of N₂ into CH₄-rich ice could be much less.”

This refers to a ternary phase diagram in Tan & Kargel, 2017, and is Figure 1 below (with annotated additions by me). As “DETAILS” show clearly, at 45 K the methane-rich phase in the binary N₂-CH₄ system (Figures 1 and 2 below) is nearly pure methane; it does not contain 4% nitrogen.

Thank you for raising this point and for sending the figures with the diagrams. We agree that this is an error in the text. 4% was a value read at 45 K on previous solid-liquid N₂-CH₄ diagrams in Trafton 2015 and Protopapa et al., 2017 and corresponds to the saturated amount of N₂ in CH₄ when both saturated solid phases are present (e.g. see Fig 7 of Protopapa et al., 2017).

We agree that the correct value for CH₄-rich ice at 45 K is actually <<1% as shown by your figure.

In the end, our model’s assumption is consistent with the value suggested by your diagram (<<1% N₂ in CH₄-rich ice) as we treat CH₄-rich ice as pure CH₄ ice.

This is seen also in the ternary diagram at 41.5 K (Figure 1; it gets only purer as temperature increases). Furthermore, at 45 K, the ternary system does not include three phases; it only includes two phases— methane-rich (nearly pure CH₄) solid and N₂-rich vapor. This pair of sentences is completely wrong. It must be changed. I recommend this language: “At the temperature of the CH₄ deposits modelled in this paper (~45 K, prevailing at high altitudes), the ternary phase equilibria shown by Tan and Kargel predict that two phases coexist: a very nearly pure CH₄ solid (<<1% impurities of N₂ and CO in solid solution) and N₂-rich vapor. Both the ternary and the binary systems shown by Tan and Kargel correspond closely to what has been observed by New Horizons and is consistent with what we report here in Figure 1. At lower temperatures at and below 36.9K (cold conditions prevailing in the lower elevations) the occurrence of a nitrogen-rich phase, according to the phase equilibria provided by Tan and Kargel13 also is like what New Horizons revealed in places like Sputnik Planitia; the methane-rich solid at those cold temperatures can contain much more impurities of N₂ and CO (up to a maximum of 4.6 molar % N₂ in solid solution at 36.9K). The impure (solid solution) N₂-bearing methane-dominated phase can coexist with vapor and an N₂-dominated phase below 36.9K13.”

We agree that the sentence was not well written, erroneous and confusing. We corrected it by adopting your suggestion. The full paragraph now reads:

On Pluto's surface, the volatile ices should form solid solutions whose phases follow ternary phase equilibria; they do not exhibit ideal behaviour^{13,24}. We note that sophisticated equations of state exist for the N₂-CH₄-CO system under Pluto surface conditions (CRYOCHEM¹³) and have been applied to Pluto's surface ices and atmospheric vapor, though not in a GCM. At the temperature of the CH₄ deposits modelled in this paper (~45 K, prevailing at high altitudes), the ternary phase equilibria shown by Tan and Kargel¹³ predict that two phases coexist: a very nearly pure CH₄ solid (<<1% impurities of N₂ and CO in solid solution) and N₂-rich vapor. Both the ternary and the binary (N₂-CH₄) systems are consistent to first order with the observations by New Horizons and with what we report in this paper and in Figure 1.

However, the ternary and binary systems cannot yet fully explain the great diversity and striking spatial heterogeneity of ice mixtures observed on Pluto's surface (CH₄-rich, N₂-rich and N₂-rich:CH₄-rich) under the relatively similar surface pressure and almost pure N₂ atmosphere across the globe. For instance, the dilution of CH₄ in the N₂-rich terrains of Sputnik Planitia is only ~0.3 to 0.5%, rather than the ~4% expected for 3-phase equilibrium. We interpret Pluto to be a non-equilibrium dynamical environment with continuous exchange of materials (condensation, sublimation, atmospheric transport). In this context, it remains unclear whether the ternary phase equilibrium plays a significant role in controlling the partial pressure and ice mixing ratios on Pluto. Future work involving laboratory experiments, spectroscopic analyses, thermodynamic models and GCMs is strongly needed to improve the models, constrain the timescales for ice relaxation toward thermodynamics equilibrium and explore in detail the effect of the ternary phase equilibrium on Pluto (and on other Trans-Neptunian objects).

Here, for simplicity in coding with a GCM, we have adopted Raoult's Law, as in previous GCM studies^{7,16,17}. We consider the mixtures N₂:CH₄ and N₂:CO with 0.5% of CH₄ and 0.3% of CO respectively, as retrieved from telescopic observations and from New Horizons observations^{4,25}. However, we note that Raoult's Law was at first intended for vapor-liquid equilibria and ideal solutions only, which do not include the N₂-CH₄-CO system observed on Pluto. Despite the fact that this approximation gives good results and allows for reproducing the atmospheric mixing ratios observed by New Horizons observations¹⁷, it may still introduce errors on sublimation and condensation rates of the different types of ice. This approximation seems sufficient for our present needs and for the study of CH₄-rich ice deposits on Pluto's surface, which are the focus of this paper, but we acknowledge that it leads to some unevaluated uncertainties in CH₄ solid phase stability.

Hopefully we address with this new paragraph the reviewer's comments. We have the feeling that the combination between the GCM (and the dynamical processes, at diurnal, seasonal and astronomical timescales) and CRYOCHEM (and the relaxation toward equilibrium) would be a powerful tool in the future that could reconcile the different observations that remain unexplained to date (e.g., while the compositional maps of Protopapa et al., 2017 reveal the presence of three latitudinal bands on Pluto's encounter hemisphere, the GCM reproduces only

the first two) and better understand how the complex cryogenic worlds of Pluto and other KBOs work.

Lines 390-394 also make a gratuitous statement about uncertainties. The manuscript says: “Here, for simplicity in coding with a GCM, we have adopted Raoult’s Law, as in previous GCM studies^{7,16,17}. This approximation is sufficient for our present needs and for the study of CH₄-rich ice deposits on Pluto’s surface, which are the focus of this paper, but we acknowledge that it leads to some uncertainties of few percent in the computation of the CH₄ sublimation and condensation rates.”

Given the mistakes made in lines 387-390, it is clear that the authors have actually not evaluated what errors may have been introduced by a reliance on Raoult’s Law or how other/better equations of state could affect the model output. The errors are potentially impactful on the major qualitative conclusions of this paper, not just on a few percent error in deposition and sublimation rates. It may affect the stability and mapped distribution of volatiles and the need for a methane-rich atmosphere at higher elevations.

My deference to the authors is because I also lack my own full analysis of Raoult’s Law vs. better thermodynamics, but I make clear that no other science field to my knowledge— certainly not Chemistry— allows an application of Raoult’s Law to volatile solids’ equilibria with vapor. This selection of statements in lines 390-394 must be rephrased. I suggest this: “Here, for simplicity in coding with a GCM, we have adopted Raoult’s Law, as in previous GCM studies^{7,16,17}. This approximation seems sufficient for our present needs and for the study of CH₄-rich ice deposits on Pluto’s surface, which are the focus of this paper, but we acknowledge that it leads to some unevaluated uncertainties in CH₄ solid phase stability.”

OK we agree with this comment. We corrected as suggested.

--Jeffrey S. Kargel

We thank the reviewer again for providing many details about the ternary phase equilibria, which are very useful and motivate us to improve the GCM by coupling it with CRYOCHEM in the future.

We note that there is a global lack of surface temperature measurements on Pluto, that prevents us for understanding how the ices evolve and what their thermodynamic state is. Future observations in the mid-IR with large telescopes may help providing additional constraints on surface temperatures. For now, estimates of surface temperatures actually come from thermal models and GCMs.

REVIEWERS' COMMENTS:

Reviewer #2 (Remarks to the Author):

Final review

I recommend acceptance, but allow the authors to make further changes at their discretion, along the following lines. The authors may see below that I attempt to give a rationale for using Raoult's Law. (But I don't advise keeping that assumption in future work.) The manuscript is much improved, with the major errors corrected, and the rest of the manuscript, as before, being acceptable and in most parts it is really amazing and intellectually stimulating. My remaining comments below pertain only to the revised language of that one paragraph, now numbered lines 399-419. I offer suggestions that should be optional for the authors to implement, but they should consider them. I say optional, because I don't want the revision and review to go into an endless loop; up until now, however, the process has been crucially important and helpful to the manuscript. The suggestions below would bring the manuscript into alignment with phase equilibria as they are known and understood now (there always is big room for improvement). For these further comments, it is best to refer to Figure 3 of my previous review.

Optional change #1, lines 399-401, which currently read "However, the ternary and binary systems cannot yet fully explain the great diversity and striking spatial heterogeneity of ice mixtures observed on Pluto's surface (CH₄-rich, N₂-rich and N₂-rich:CH₄-rich) under the relatively similar surface pressure and almost pure N₂ atmosphere across the globe."

I refer the authors to Figure 3 that was sent before.

In order to have Sputnik Planitia be composed of a single phase of nitrogen containing 0.3-0.5% CH₄ in solid solution, the bulk Sputnik Planitia ice must reside in the green phase field. (And don't believe the phase diagram below the alpha-beta transition, which is not represented, at around 35.4 K as I recall.) So composition U, for instance, would be around 2% CH₄ in solid solution with N₂ (just eyeballing the figure). There is no problem to have 0.3-0.5% CH₄ in 99.5% N₂ ice (phase S1) is the green phase field. 0.3% CH₄ is solidly in the green phase field. Whether 0.3% CH₄ or 2% as the dashed vertical red line is drawn, it is in the S1 phase field and the potential for a separate CH₄-rich phase is eliminated, since these compositions are left of (N₂-richer than) the solvus curve. So there is only one solid, N₂-rich phase S1. If this phase warms up to temperature W, the nitrogen sublimates into a nearly pure N₂ vapor, and almost all of the minor impurity of methane then forms its own S2 phase, which at that temperature has around 4% N₂ in it. The rest of the N₂ turns into vapor, and a bare

trace (invisible in the figure) of CH₄ also is in the vapor. After all the solid nitrogen S1 phase completely sublimates away, the temperature can rise, and as it does, the methane phase S2 becomes purer, and the trace of methane in the vapor also increases (but it is not a visible trace in the phase diagram as published until above temperature Y. If we had better resolution in the phase diagram, we would see that a system with a bulk CH₄ content of 0.3-0.5% would have a decreasing quantity of phase S2 (which also keeps getting purer as temperature warms), and the solid S2 phase also would go completely into vapor at some temperature below Y, maybe around 40 K (I guess, because the phase diagram loses resolution of that trace of methane in the vapor). If you want a methane solid phase S2 to remain in some quantity until 44K, you need a bulk composition of about 2% methane (the vertical red dashed line). If there are any N₂-ice deposits that have large amounts of methane solids, then you need a bulk composition that lies in the yellow field. In this case, not only can you have a separate phase S2, but the nitrogen phase S1 also is CH₄-rich; that's when you get the 4% CH₄ dissolved in N₂ ice phase S1.

So the full range of compositions mentioned in the revised paper actually is well represented in this binary phase diagram (neglecting poor old CO, which actually also is a player). But the constraints then are that the surface is heterogeneous beyond what a single bulk composition can produce. The 0.3-0.5% CH₄ + 99.5% N₂ ice that can be present only at low temperatures CAN produce a small amount of nearly pure methane ice (but that small amount can completely cover a surface as a lag deposit) by warming to >40K. But if you have rich admixtures of CH₄ and N₂, then that bulk composition has to exist in the solvus region (yellow field). So we would be talking about two different bulk compositions (at least). Then again, Earth has basalt and granite, and it has halite evaporites and epsomite evaporites, acid brine pools and alkaline brine pools. So we must be talking about a chemically diverse Pluto. It is possible to create scenarios entirely in this phase diagram that make all of these mixtures starting with any bulk composition, but they are multi-step fractionations. That is not the same as concluding that you can't explain the materials in the phase equilibrium diagram, and that phase equilibria might not be playing a role. To say so would be like saying that Earth's water vapor saturation pressure and relative humidity might not play a role in cloud formation, rain, snow, and evaporation. Of course the phase equilibria are active on Pluto. But also, there can be divergences from equilibrium. Since the phase diagram ultimately is rooted in lab observations by impatient and short-lived humans, and Pluto's year and seasons is longer than a human career or lab experiment, it would be extremely improbable that phase equilibria is not having a commanding presence in the goings-on at Pluto.

So what to do? It would be more accurate to say: "The ternary and binary systems, as currently understood [Tan and Kargel 2017 reference here] are able to explain a great diversity of phases (CH₄-rich, N₂-rich and N₂-rich+CH₄-rich solids) that are seen on Pluto within the range of temperatures and relatively unvarying surface pressure and strongly N₂-dominated vapor composition seen on Pluto. However, the phase equilibria have not yet been applied to the specific distribution of ices and temperatures seen on Pluto or in a GCM, so our work is an attempt to fill that gap."

Lines 401-403 reads: "For instance, the dilution of CH₄ in the N₂-rich terrains of Sputnik Planitia is only ~0.3 to 0.5%, rather than the ~4% expected for 3-phase equilibrium."

I would delete this sentence. I just pointed out that if the bulk composition is in the green field, it allows 0.3% methane in solid solution with N₂; and that composition still allows evolution to a pure-ish methane phase at 40 K. If you want to keep this sentence roughly, you could say: "The dilution of CH₄ in the N₂-rich terrains of Sputnik Planitia is only ~0.3 to 0.5%, so that involves occurrence in the "S1" phase field represented in Figure 2b of Tan and Kargel (2013). That composition can evolve to nearly pure methane ice and N₂-rich vapor by warming to 40K. However, highly enriched admixtures of methane ice and nitrogen ice would require occurrence within the solvus region designated S1S2. From this, we may conclude that Titan has a compositionally heterogeneous set of mixtures. As these equations of state have not been coded for use in a GCM, we have substituted the alternative of relying on Raoult's Law, as described next. It is possible that for conditions of rapid frost deposition, the solid condensates may be amorphous mixtures, which might tend to exhibit thermodynamics somewhat like those implied by Raoult's Law. Accordingly, we interpret Pluto to be a non-equilibrium dynamical environment with continuous exchange of materials (condensation, sublimation, atmospheric transport, including on daily timescales where departures from equilibrium could be likely."

Lines 405-406 says: "In this context, it remains unclear whether the ternary phase equilibrium plays a significant role in controlling the partial pressure and ice mixing ratios on Pluto."

I would recommend removing this. It is virtually impossible that this is the case overall; transiently, perhaps. There can be transient departures from equilibrium, but just consider the length of Pluto's year and seasons... far longer than any lab rat's patience or career longevity. I cannot conceive that a very close approach to equilibrium is not attained overall. I would not say the same about the nightly and daily transitions, because that is comparable to lab experiments, and maybe kinetics plays a big factor.

The ending of the paragraph starting from Line 406 "Future work..." is fine.

REVIEWERS' COMMENTS:

Reviewer #2 (Remarks to the Author):

Final review

I recommend acceptance, but allow the authors to make further changes at their discretion, along the following lines. The authors may see below that I attempt to give a rationale for using Raoult's Law. (But I don't advise keeping that assumption in future work.) The manuscript is much improved, with the major errors corrected, and the rest of the manuscript, as before, being acceptable and in most parts it is really amazing and intellectually stimulating. My remaining comments below pertain only to the revised language of that one paragraph, now numbered lines 399-419. I offer suggestions that should be optional for the authors to implement, but they should consider them. I say optional, because I don't want the revision and review to go into an endless loop; up until now, however, the process has been crucially important and helpful to the manuscript. The suggestions below would bring the manuscript into alignment with phase equilibria as they are known and understood now (there always is big room for improvement). For these further comments, it is best to refer to Figure 3 of my previous review.

Optional change #1, lines 399-401, which currently read "However, the ternary and binary systems cannot yet fully explain the great diversity and striking spatial heterogeneity of ice mixtures observed on Pluto's surface (CH₄-rich, N₂-rich and N₂-rich:CH₄-rich) under the relatively similar surface pressure and almost pure N₂ atmosphere across the globe."

I refer the authors to Figure 3 that was sent before.

In order to have Sputnik Planitia be composed of a single phase of nitrogen containing 0.3-0.5% CH₄ in solid solution, the bulk Sputnik Planitia ice must reside in the green phase field. (And don't believe the phase diagram below the alpha-beta transition, which is not represented, at around 35.4 K as I recall.) So composition U, for instance, would be around 2% CH₄ in solid solution with N₂ (just eye-balling the figure). There is no problem to have 0.3-0.5% CH₄ in 99.5% N₂ ice (phase S1) is the green phase field. 0.3% CH₄ is solidly in the green phase field. Whether 0.3% CH₄ or 2% as the dashed vertical red line is drawn, it is in the S1 phase field and the potential for a separate CH₄-rich phase is eliminated, since these compositions are left of (N₂-richer than) the solvus curve. So there is only one solid, N₂-rich phase S1. If this phase warms up to temperature W, the nitrogen sublimates into a nearly pure N₂ vapor, and almost all of the minor impurity of methane then forms its own S2 phase, which at that temperature has around 4% N₂ in it. The rest of the N₂ turns into vapor, and a bare trace (invisible in the figure) of CH₄ also is in the vapor. After all the solid nitrogen S1 phase completely sublimates away, the temperature can rise, and as it does, the methane phase S2 becomes purer, and the trace of methane in the vapor also increases (but it is not a visible trace in the phase diagram as published until above temperature Y. If we had better resolution in the phase diagram, we would see that a system with a bulk CH₄ content of 0.3-0.5% would have a decreasing quantity of phase S2 (which also keeps getting purer as temperature warms), and the solid S2 phase also would go completely into vapor at some

temperature below Y, maybe around 40 K (I guess, because the phase diagram loses resolution of that trace of methane in the vapor). If you want a methane solid phase S2 to remain in some quantity until 44K, you need a bulk composition of about 2% methane (the vertical red dashed line). If there are any N2-ice deposits that have large amounts of methane solids, then you need a bulk composition that lies in the yellow field. In this case, not only can you have a separate phase S2, but the nitrogen phase S1 also is CH4-rich; that's when you get the 4% CH4 dissolved in N2 ice phase S1.

So the full range of compositions mentioned in the revised paper actually is well represented in this binary phase diagram (neglecting poor old CO, which actually also is a player). But the constraints then are that the surface is heterogeneous beyond what a single bulk composition can produce. The 0.3-0.5% CH4 + 99.5% N2 ice that can be present only at low temperatures CAN produce a small amount of nearly pure methane ice (but that small amount can completely cover a surface as a lag deposit) by warming to >40K. But if you have rich admixtures of CH4 and N2, then that bulk composition has to exist in the solvus region (yellow field). So we would be talking about two different bulk compositions (at least). Then again, Earth has basalt and granite, and it has halite evaporites and epsomite evaporites, acid brine pools and alkaline brine pools. So we must be talking about a chemically diverse Pluto. It is possible to create scenarios entirely in this phase diagram that make all of

these mixtures starting with any bulk composition, but they are multi-step fractionations. That is not the same as concluding that you can't explain the materials in the phase equilibrium diagram, and that phase equilibria might not be playing a role. To say so would be like saying that Earth's water vapor saturation pressure and relative humidity might not play a role in cloud formation, rain, snow, and evaporation. Of course the phase equilibria are active on Pluto. But also, there can be divergences from equilibrium. Since the phase diagram ultimately is rooted in lab observations by impatient and short-lived humans, and Pluto's year and seasons is longer than a human career or lab experiment, it would be extremely improbable that phase equilibria is not having a commanding presence in the goings-on at Pluto.

We agree with these comments.

So what to do? It would be more accurate to say: “

Lines 401-403 reads: “For instance, the dilution of CH4 in the N2-rich terrains of Sputnik Planitia is only ~0.3 to 0.5%, rather than the ~4% expected for 3-phase equilibrium.”

I would delete this sentence. I just pointed out that if the bulk composition is in the green field, it allows 0.3% methane in solid solution with N2; and that composition still allows evolution to a pure-ish methane phase at 40 K. If you want to keep this sentence roughly, you could say: “The dilution of CH4 in the N2-rich terrains of Sputnik Planitia is only ~0.3 to 0.5%, so that involves occurrence in the “S1” phase field represented in Figure 2b of Tan and Kargel (2013). That composition can evolve to nearly pure methane ice and N2-rich vapor by warming to 40K. However, highly enriched admixtures of methane ice and nitrogen ice would require occurrence within the solvus region designated S1S2. From this, we may conclude that Titan has a compositionally heterogeneous set of mixtures. As these equations of state have not been coded for use in a GCM, we have substituted the alternative of relying on Raoult's Law, as described next. It is possible that for conditions of rapid frost deposition, the solid condensates may be amorphous mixtures, which might tend to exhibit thermodynamics somewhat like those implied

by Raoult's Law. Accordingly, we interpret Pluto to be a non-equilibrium dynamical environment with continuous exchange of materials (condensation, sublimation, atmospheric transport, including on daily timescales where departures from equilibrium could be likely."

Done.

Lines 405-406 says: "In this context, it remains unclear whether the ternary phase equilibrium plays a significant role in controlling the partial pressure and ice mixing ratios on Pluto." I would recommend removing this. It is virtually impossible that this is the case overall; transiently, perhaps. There can be transient departures from equilibrium, but just consider the length of Pluto's year and seasons... far longer than any lab rat's patience or career longevity. I cannot conceive that a very close approach to equilibrium is not attained overall. I would not say the same about the nightly and daily transitions, because that is comparable to lab experiments, and maybe kinetics plays a big factor.

The ending of the paragraph starting from Line 406 "Future work..." is fine.

Done.

We thank the reviewer for these detailed suggestions.